# FINE-TUNING MULTIMODAL LLMS TO FOLLOW ZERO-SHOT DEMONSTRATIVE INSTRUCTIONS

**Juncheng Li**[1,2]* **Kaihang Pan**[1]* **Zhiqi Ge**[1]* **Minghe Gao**[1]* **Wei Ji**[2]  **Wenqiao Zhang**[1]
**Tat-Seng Chua**[2]  **Siliang Tang**[1]†  **Hanwang Zhang**[3]  **Yueting Zhuang**[1]†
[1]Zhejiang University,  [2]National University of Singapore,  [3]Nanyang Technological University

## ABSTRACT

Recent advancements in Multimodal Large Language Models (MLLMs) have been utilizing Visual Prompt Generators (VPGs) to convert visual features into tokens that LLMs can recognize. This is achieved by training the VPGs on millions of image-caption pairs, where the VPG-generated tokens of images are fed into a frozen LLM to generate the corresponding captions. However, this image-captioning based training objective inherently biases the VPG to concentrate solely on the primary visual contents sufficient for caption generation, often neglecting other visual details. This shortcoming results in MLLMs' underperformance in comprehending demonstrative instructions consisting of multiple, interleaved, and multimodal instructions that demonstrate the required context to complete a task. To address this issue, we introduce a generic and lightweight Visual Prompt Generator Complete module (`VPG-C`), which can infer and complete the missing details essential for comprehending demonstrative instructions. Further, we propose a synthetic discriminative training strategy to fine-tune `VPG-C`, eliminating the need for supervised demonstrative instructions. As for evaluation, we build `DEMON`, a comprehensive benchmark for demonstrative instruction understanding. Synthetically trained with the proposed strategy, `VPG-C` achieves significantly stronger zero-shot performance across all tasks of `DEMON`. Further evaluation on the MME and OwlEval benchmarks also demonstrate the superiority of `VPG-C`. The code and models are available at `https://github.com/DCDmllm/Cheetah`.

## 1 INTRODUCTION

Recent advances in Multimodal Large Language Models (MLLMs) (Li et al., 2023c; Liu et al., 2023; Zhu et al., 2023a) have exhibited promising capabilities in processing single-image instructions, such as producing detailed image descriptions and answering questions about the image. However, they fall short in **demonstrative instructions** consisting of multiple, interleaved, and multimodal instructions that demonstrate the required context to complete a task. For instance, the instruction in Figure 1 contains interleaved visual and textual context, requiring the model to determine the authenticity of the milk in the second image based on the official image provided in the first.

An MLLM should at least have the following two capabilities to comprehend demonstrative instructions effectively:

**1) Not just the primary subject:** Beyond focusing on the primary visual content, it should be able to meticulously discern the details within the demonstrations. These details, complementing the primary content, play a crucial role in semantically connecting the instructions. A case in point is Figure 1, wherein accurate discernment relies on recognizing the logo detail on a milk carton.

**2) Reasoning-aware details:** How to decide what details are complementary to the reasoning? We expect that an MLLM may "think twice", that is, given a preliminary reasoning using the primary contents, it would know what additional contents are needed as complementary details. For example, in Figure 1, after preliminary reasoning, the model should re-attend details such as the logo and

---

*Equal Contribution. †Corresponding Authors.

**Demonstrative Instruction**

*Compared to the genuine milk displayed in* [image] *, please examine the packaging design of my purchased milk shown in* [image] *. Is the purchased milk genuine?*

**InstructBLIP-Answer:** *Yes, the purchased milk is genuine.*

Visualized attention maps of the InstructBLIP's VPG.

**Ours-Answer:** *No, the cow icon within the red diamond-shaped box on the purchased milk packaging is noticeably different from that of the genuine milk.*

Stage1: initial attention maps    Stage2: attention maps after preliminary reasoning

Figure 1: An example of InstructBLIP (Dai et al., 2023) and our MLLM enhanced by `VPG-C`.

brand name on the milk carton, thereby discerning its authenticity. However, to follow zero-shot demonstrative instructions, this "reasoning-aware" capability should be acquired without the need for supervised demonstrative instructions.

Unfortunately, we find that the reason why existing MLLMs are not effective in demonstrative instructions is due to the lack of the above capabilities. More specifically, the crux lies in the Visual Prompt Generator (VPG) in MLLMs. VPG, such as Q-former (Li et al., 2023c) and Resampler (Alayrac et al., 2022), translates visual features into tokens recognizable by LLMs, and the translation is trained on millions of image-caption pairs by feeding the VPG-generated tokens of images into a frozen LLM which generates the corresponding captions. However, this image captioning training strategy inevitably introduces the inductive bias that VPG only focuses on the primary visual contents which are just enough for the captioning task, but tends to omit other visual details. For example in Figure 1, the averaged attention map of InstructBLIP (Dai et al., 2023) (Figure 1) shows a dominant focus on the primary contents, neglecting the logo detail, which is however the key to answering the question.

To this end, we propose a lightweight Visual Prompt Generator Complete module (`VPG-C`), which can infer and complete the missing details essential for comprehending demonstrative instructions (Section 2.1). As shown Figure 2, 1) `VPG-C` first derives the instruction-specific guidance by intercepting the intermediate LLM's output of the primary contents extracted by a conventional VPG, and then

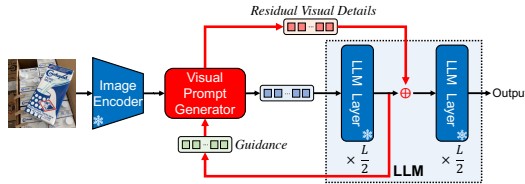

Figure 2: An overview of `VPG-C`.

2) guides the VPG to recover the missing visual residual details. Finally, 3) these residual details are then seamlessly reintegrated into the intermediate LLM's layer via a skip connection. Together with the original intermediate output, `VPG-C` is expected to provide an improved comprehension of the demonstration instructions. Yet, `VPG-C` is not ready to follow zero-shot demonstrative instructions because the "Guide" step requires fine-tuning to specialize in missing detail recovery. Therefore, we propose a synthetic discriminative training strategy to fine-tune VPG-C, without the need for the expensive data collection of "detail-caption" pairs (Section 2.2).

To evaluate `VPG-C` and diagnose existing MLLMs, we build `DEMON`, a comprehensive benchmark for demonstrative instruction understanding, covering 31 diverse tasks across 7 categories, as shown in Figure 4 (Section 3). Systematic evaluation on `DEMON` confirms the limitation of existing MLLMs in demonstrative instructions. Without additional demonstrative instruction data, the lightweight `VPG-C` module can be effectively tuned by the synthetic training strategy in several hours with a single A100 GPU. While computation- and data- efficient, `VPG-C` significantly outperforms existing MLLMs on the `DEMON` benchmark. Zero-shot evaluation on other multimodal instruction benchmarks (Fu et al., 2023; Ye et al., 2023) also indicates considerable improvement by `VPG-C`.

## 2 METHOD

### 2.1 VISUAL PROMPT GENERATOR COMPLETE

As illustrated in Figure 2, `VPG-C` is built upon the frozen LLM (Vicuna-7B (Chiang et al., 2023)) and vision encoder (EVA-CLIP (Fang et al., 2023)). We adopt the widely used Q-Former from BLIP-

2 (Li et al., 2023c) as our visual prompt generator. `VPG-C` first uses the intermediate output of the LLM to infer instruction-specific guidance. This then assists the VPG in attending to the missing visual details from the images. By merging these residual details back via a skip connection, `VPG-C` achieves a thorough grasp of the demonstrative instruction.

Given a demonstrative instruction, we first adopt the Q-former to generate general visual prompts for each image in the instruction. Q-former takes a fixed number of $K$ query vectors to interact with image features by several cross-attention layers, and the output query representations are used as visual prompts, which are inserted into the position of their corresponding images in the instruction. We denote the input instruction for the language decoder as $\mathcal{H}^0 = \{\mathbf{h}_1^0, \mathbf{h}_2^0, ..., \mathbf{v}_{11}^0, ..., \mathbf{v}_{1K}^0, ..., \mathbf{h}_i^0, ..., \mathbf{v}_{j1}^0, ..., \mathbf{v}_{jK}^0, ..., \mathbf{h}_N^0\}$, where $\mathbf{h}_i^0$ represents the $i$-th text token and $\mathcal{V}_j^0 = \{\mathbf{v}_{j1}^0, ..., \mathbf{v}_{jK}^0\}$ represents the $K$ visual prompts for the $j$-th interleaved image. Taking the instruction as input to the $L$-layer language decoder, we then extract the hidden representation of the last input token $\mathbf{h}_N^{L/2}$ at the $\frac{L}{2}$-th layer, which can fully perceive the whole multimodal context during the first $\frac{L}{2}$ layers and contains comprehensive instruction-aware semantics. Next, we infer the instruction-specific guidance $\mathbf{g}$ from $\mathbf{h}_N^{L/2}$ via a linear projection layer: $\mathbf{g} = \mathbf{Linear}(\mathbf{h}_N^{L/2})$.

After obtaining the instruction-specific guidance from the language decoder, we compose it with a new set of learnable queries: $\mathbf{g} + \mathcal{Q}$, where $\mathcal{Q} \in \mathbf{R}^{K \times d}$ and $\mathbf{g}$ is added to each query of $\mathcal{Q}$. Then, we reuse the same Q-former with the above conditionally generated queries to attend to residual visual details, thus obtaining the visual prompts $\overline{\mathcal{V}}_j = \{\overline{\mathbf{v}}_{j1}, ..., \overline{\mathbf{v}}_{jK}\}$ for each image $j$, which contains the complementary details missed by the original visual prompts. Finally, the transformed $\overline{\mathcal{V}}_j$ is reintegrated with the corresponding original intermediate representations of $\mathcal{V}_j^{L/2}$, via skip connection: $\tilde{\mathcal{V}}_j^{L/2} = \mathcal{V}_j^{L/2} + \mathbf{Linear}(\overline{\mathcal{V}}_j)$, which is taken as the input to the $(\frac{L}{2} + 1)$-th layer.

**Efficient training.** Our `VPG-C` module is parameter-efficient as the Q-former is frozen and only a set of query embeddings and two linear projection layers need to be fine-tuned, which only account for **0.09% (∼6.3M)** of the entire model. To stabilize the training process (Zhang & Agrawala, 2023), we initialize the linear projection layers with zeros. Thus, at the early training stage, the input to the $(\frac{L}{2} + 1)$-th layer can be converted to: $\tilde{\mathcal{V}}_j^{L/2} = \mathcal{V}_j^{L/2}$, which will not cause any influence on LLMs.

**Analysis on inserting `VPG-C` in the intermediate layer ($\frac{L}{2}$): 1) Guidance generation.** Previous studies have shown that features provided by the intermediate layer may suffice to preliminarily understand the given input samples (Xin et al., 2020) and can serve as guidance hints to improve training (Romero et al., 2014). Thus, generating guidance in the intermediate layer allows the model to form a preliminary understanding of the given instruction. Generating guidance too early could be problematic, as the model might not have gathered sufficient contextual information to generate effective guidance. Conversely, generating guidance too late could result in the model's attention being narrowly focused on what it perceives as the final answer, hindering its ability to guide the Q-former to extract relevant details from the images. Therefore, placing the guidance generation step in the intermediate layer strikes a balance. **2) Detail reintegration.** Intermediate-layer reintegration of residual visual details preserves prior knowledge and allows subsequent layers to integrate new information effectively. Reintegrating residual details too early in the pipeline might overwrite important context, while reintegrating it too late could limit the impact on the model's reasoning. Therefore, the intermediate layer offers a strategic position for residual details reintegration, enabling the model to reason effectively and arrive at the correct answers by leveraging the complemented visual residual details. We further provide **quantitative analysis** in Section 4.4.

## 2.2 SYNTHETIC DISCRIMINATIVE TRAINING STRATEGY

The proposed training strategy diagnoses the areas initially ignored by Q-former according to its cross-attention maps between the queries and the image features, and generates a synthetic image by performing several types of editing on the ignored areas. Then, an inter-image discriminative task is formulated as describing the subtle difference between the original and the synthetic images. Considering the edits are performed in the mostly ignored areas, `VPG-C` is challenged to recover the missing details to describe the difference. An overview is illustrated in Figure 3.

**Editing target identification.** The Q-former takes the queries to interact with frozen image features through several cross-attention layers and uses the output query representations as the visual

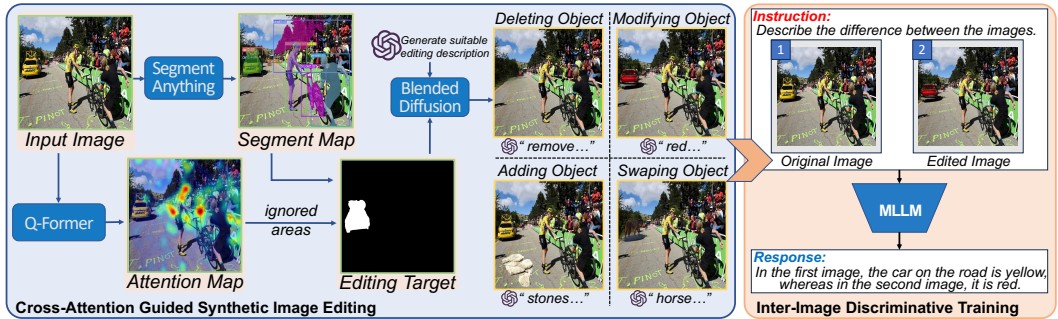

Figure 3: Pipeline demonstration of synthetic discriminative training strategy for `VPG-C`.

prompts. Therefore, the cross-attention maps between queries and image features reflect the interest of queries. We average the cross-attention maps across all layers and all queries to obtain the global cross-attention map $\mathcal{A}$, where the value $\mathcal{A}_{ij}$ indicates the significance degree of the corresponding image feature by the original task-agnostic Q-former queries. After that, we employ the advanced vision foundation model (Kirillov et al., 2023) to obtain all the objects with segmentation masks in the image. Then, the significance degree of each object $\Phi(o_i)$ is computed based on the cross-attention map $\mathcal{A}$ with RoIAlign (He et al., 2017), where we average the values of $\mathcal{A}$ within the mask $m_i$ by interpolation. $\Phi(o_i)$ reflects what degree the visual features of object $o_i$ is extracted by the Q-former. Thus, we select the most ignored objects based on the $\Phi(o_i)$ value.

**Editing description generation.** We define four types of editing: *modifying objects, swapping objects, deleting objects, and adding objects.* Given the selected object, we instruct ChatGPT (OpenAI, 2023a) to generate a suitable editing description that is in harmony with the context, where ChatGPT is prompted with the corresponding image caption and detailed object information (*i.e.,* labels, positions). For adding objects, we only select `BACKGROUND` objects to add objects.

**Synthetic image generation.** After obtaining the editing description, we generate the synthetic image using a text-to-image latent diffusion model (*i.e.,* Blended Diffusion (Avrahami et al., 2022)). Blended Diffusion performs local editing on the image according to the target object mask and the editing description, thus rendering the synthetic image. To ensure quality, we filter the edited images using CLIP similarity (Radford et al., 2021b).

**Inter-image discriminative training.** Given the original image and the synthetic image pair, along with the task instruction (*"Describe the difference between the images"*), the inter-image discriminative training task is defined as generating sentences to describe the subtle difference between the images. We convert the editing description to acquire the ground-truth sentences.

## 3 DEMON BENCHMARK

**Data format.** All task instances are transformed into a unified instruction-response form for zero-shot evaluation. Formally, each instance in `DEMON` consists of the following components:

- `Task_Instruction`: provides a complete natural language definition of a given task, including the input/output format and the task objective.
- `Task_Instance`: is a concrete instance of a given task that consists of demonstrative image-text sequential context (*e.g.*, visually-rich textbooks, specific questions about the context).
- `Response`: represents the target output in natural language for a given task instruction and task instance. For classification tasks, we convert the class labels as options into the instruction and ask the model to output the option index in natural language as the response.

Without any specific emphasis, we use the term "instruction" to refer to the combination of `Task_Instruction` and `Task_Instance`. For each task, we manually design 10 `Task_Instruction` templates in natural language to increase the diversity.

**Task collection and categorization.** To comprehensively benchmark the demonstrative instruction following ability, we extensively gather a wide variety of multimodal datasets from different fields and scenarios. As illustrated in Figure 4, `DEMON` has three important properties: **1) Demonstrative vision-language context,** all the instructions contain sequences of inter-related images and texts, such as storyboards with scripts, and textbooks with diagrams. **2) Diverse forms of complex**

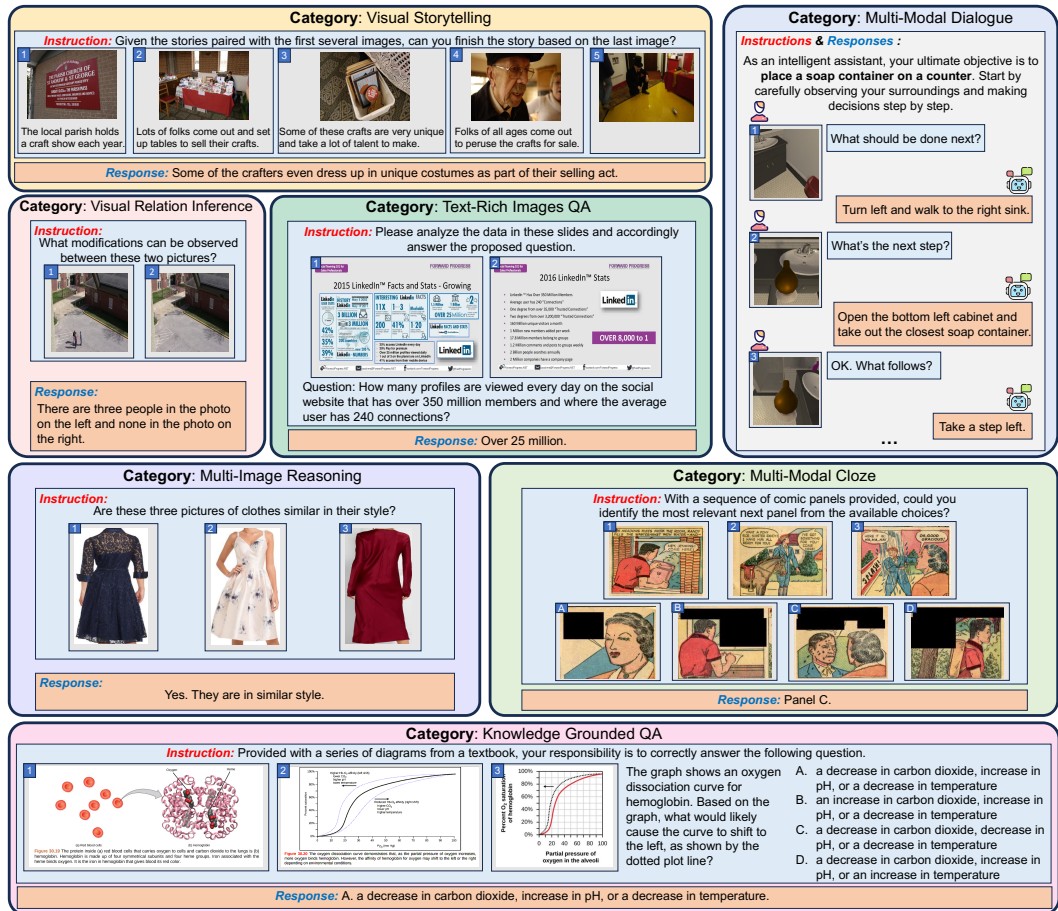

Figure 4: Demonstrations and task taxonomy of the proposed DEMON benchmark.

**instructions,** the instructions range from designing panels for comics, to discovering differences between surveillance images, and to conversational embodied tasks. **3) Vast range of instruction-following scenarios,** the benchmark covers multiple practical scenarios, including cartoons, industrial visuals, driving recordings, recipes, *etc*.

**Evaluation protocols.** Thanks to the unified task format of DEMON, all tasks can be evaluated in a zero-shot manner. For the open-ended generation tasks, we adopt *ROUGE-L* for evaluation. For the tasks that require the models to output option indexes, we take *Accuracy* as the evaluation metric. While well-formated options are provided, we empirically observe that many MLLMs struggle to strictly follow instructions to output the option indexes but generate free-form text. Thus, when models do not exactly output the required options, we match their outputs to one of the given options based on the TF-IDF distance, which we find is more robust than model-based methods (OpenAI, 2023a; Reimers & Gurevych, 2019). Since we explore a large number of tasks, we take a maximum of 500 instances per task for evaluation efficiency and exclude several datasets that are difficult to obtain and are subject to strict copyright restrictions (referred to as DEMON-Core). Meanwhile, we report the full version of the benchmark to facilitate future research on large-scale multimodal instruction tuning (referred to as DEMON-Full). Unless specifically stated, we use DEMON to refer to DEMON-Core in the following.

Table 1: Detailed statistics of DEMON benchmark.

|  | Tasks | Scenarios | Images | Instructions | Avg. Images / Instruction | Avg. Words / Instruction |
|---|---|---|---|---|---|---|
| DEMON-Core | 29 | 19 | 62.81K | 18.18K | 3.46 | 92.69 |
| DEMON-Full | 31 | 20 | 1.77M | 477.72K | 3.70 | 97.58 |

**Benchmark analysis.** Table 1 details the statistics. DEMON benchmark covers 31 tasks of 7 categories across 20 scenarios. In total, DEMON-Full includes 477.72K instruction-response pairs,

Table 2: Average results of zero-shot evaluation on each task category of `DEMON` Benchmark.

| | Multimodal Dialogue | Visual Storytelling | Visual Relation Inference | Multimodal Cloze | Knowledge Grounded QA | Text-Rich Images QA | Multi-Image Reasoning |
|---|---|---|---|---|---|---|---|
| BLIP-2 (Li et al., 2023c) | 26.12 | 21.31 | 10.67 | 17.94 | 39.23 | 33.53 | 39.65 |
| InstructBLIP (Dai et al., 2023) | 33.58 | 24.41 | 11.49 | 21.20 | 47.40 | 44.40 | 48.55 |
| LLaMA-Adapter V2 (Gao et al., 2023) | 14.22 | 17.57 | 13.51 | 18.00 | 44.80 | 32.00 | 44.03 |
| LLaVA (Liu et al., 2023) | 7.79 | 10.70 | 8.27 | 15.85 | 36.20 | 28.33 | 41.53 |
| MiniGPT-4 (Zhu et al., 2023a) | 13.69 | 17.07 | 7.95 | 16.60 | 30.27 | 26.40 | 43.50 |
| mPLUG-Owl (Ye et al., 2023) | 12.67 | 19.33 | 5.40 | 16.25 | 33.27 | 32.47 | 42.50 |
| OpenFlamingo (Awadalla et al., 2023) | 16.88 | 24.22 | 13.85 | 21.65 | 32.00 | 30.60 | 41.63 |
| Otter (Li et al., 2023a) | 15.37 | 15.57 | 11.39 | 16.00 | 41.67 | 27.73 | 43.85 |
| **VPG-C** | **37.50** | **25.20** | **25.90** | **22.15** | **48.60** | **44.93** | **50.28** |

serving as a large-scale benchmark for demonstrative instruction following. On average, each instruction contains 3.70 images and 97.58 words. Please refer to Appendix B for more details.

## 4 EXPERIMENTS

### 4.1 ZERO-SHOT EVALUATION ON DEMON BENCHMARK

**Comparison with advanced MLLMs.** In this section, we conduct comprehensive evaluation of our `VPG-C` and the recent advanced MLLMs on the proposed `DEMON` benchmark. For all methods, we choose versions with parameter sizes less than 10B. Please refer to Appendix D, F for details. The average results of each task category are summarized in Table 2, which indicates the following.

- `VPG-C` consistently outperforms existing models by a large margin across all categories, which demonstrates the stronger generalizability to follow such complicated demonstrative instructions.
- While previous works mainly fine-tune on massive multimodal instruction data, `VPG-C` still achieves the highest performance using synthetic training data with much lower computation cost. This validates the effectiveness of the proposed `VPG-C` module and its synthetic training strategy.
- Compared with previous works that fine-tune the large-scale language decoder or visual encoder (*i.e.,* LLaVA, mPLUG-Owl), our model only tunes the lightweight `VPG-C` module with 6.3M parameters and achieves significant performance gain.
- `VPG-C` exhibits significant superiority in several challenging tasks. For instance, `VPG-C` surpasses SOTA methods by 3.92% on multimodal dialogue, which requires models to effectively associate the interleaved images mentioned in different turns of the dialogue.

**Innovative findings.** The extensive evaluation on `DEMON` benchmark reveals several key findings.

- **Poor performance on demonstrative instructions.** While several models (*e.g.,* OpenFlamingo, Otter, mPLUG-owl) have been trained on interleaved vision-language data, such as mmc4 (Zhu et al., 2023b), they still struggle to perform well on the demonstraive instructions. We suppose that while mmc4 contains sequences of interleaved images as context, the web-crawled images are often weakly related. In contrast, the images and text in demonstrative instructions are highly related, requiring models to deeply associate them to comprehend the task intents.
- **Limited instruction following ability.** Despite existing vision-language models leveraging state-of-the-art LLMs, which have demonstrated impressive ability in following language instructions, this competence seems to falter when dealing with complex multimodal instructions. For instance, when tasked with selecting the correct answer from a choice list given the context of images and texts, we observed some models inclining more towards describing the contents of the images instead of addressing the posed questions. This is perceived as a deficiency in the image-text alignment training process, to which we attribute the discrepancy.
- **Failing to process image-choice questions.** When dealing with multimodal cloze tasks, all models are limited to processing instructions that involve images as options. We hope future work to utilize the new benchmark to make progress on this type of demonstrative instructions.

### 4.2 ZERO-SHOT EVALUATION ON MME BENCHMARK

We evaluate our `VPG-C` on the concurrently proposed MME benchmark (Fu et al., 2023) to further illustrate its strong generalizability to follow a diverse range of single-image instructions. MME benchmark measures both perception and cognition abilities on a total of 14 subtasks. We report the averaged results of perception tasks and cognition tasks in Table 3, respectively. While we

Table 3: Zero-shot evaluation of perception and cognition abilities on MME benchmark.

|  | BLIP-2 | InstructBLIP | LA-V2 | LLaVA | MiniGPT-4 | mPLUG-Owl | Otter | **VPG-C** |
|---|---|---|---|---|---|---|---|---|
| Perception | 1293.84 | 1212.82 | 972.67 | 502.82 | 866.57 | 967.34 | 1292.26 | **1299.24** |
| Cognition | 290.00 | 291.79 | 248.93 | 214.64 | 292.14 | 276.07 | 306.43 | **321.07** |

do not use massive multimodal instruction data to fine-tune VPG-C, VPG-C still achieves superior performance, compared with the supervised instruction-tuned models. This indicates our method effectively overcomes the inherent limitation of VPGs and the completed residual details are also essential for single-image instructions. Please refer to Appendix E for detailed results.

### 4.3 HUMAN EVALUATION ON GENERAL-PURPOSE LANGUAGE GENERATION

We further conduct human evaluation on the OwlEval benchmark (Ye et al., 2023), which contains 82 open-ended questions including advertisement and poem creation, diagram and flowchart comprehension, and teaching, *etc.* Specifically, we recruit 8 well-educated people to rank the randomly shuffled responses from VPG-C, MiniGPT-4, mPLUG-Owl, OpenFlamingo, and InstructBLIP. The scores range from 1 to 5 (5 means best) and are allowed to be equal for comparable instances. As shown in Figure 5, VPG-C also demonstrates better open-ended language generation ability in various practical cases.

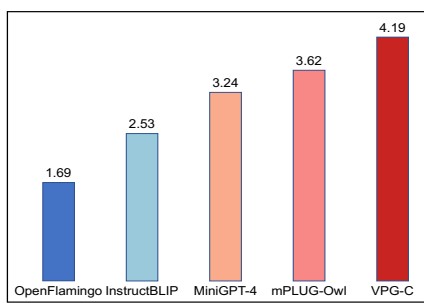

Figure 5: Human evaluation.

### 4.4 IN-DEPTH ANALYSIS

**Effectiveness of individual components.** We investigate the effectiveness of each component in Table 4. We start with the backbone model that uses the Q-former to generate visual prompts. **1)** Instead of applying VPG-C to capture missing details, we first attempt a simple heuristic-based method that directly extracts the less attended visual features according to the cross-attention maps of Q-former and reintegrates them to the intermediate layer of the LLM as ours. We fine-tune a linear layer before reintegrating with 0.5 million image-caption pairs. The results of Row 2 show that such a sample heuristic can bring some improvement. This validates *the importance of re-extracting missing visual features from images for comprehending demonstrative instructions.* **2)** Then, we replace it with VPG-C and train it only using the image-caption pairs without synthetic training. The results of Row 3 demonstrate that *VPG-C can more accurately complete the required missing details by leveraging the intermediate inference results of the LLM.* **3)** However, solely using common image-caption data can not fully unleash the power of VPG-C. Comparing Row 3 and Row 4, we observe a significant improvement for all tasks, indicating that *the proposed synthetic discriminative training strategy can methodically empower VPG-C to extract missing visual details.*

**VPG-C can better guide VPGs.** Since InstructBLIP can perform conditional visual feature extraction, we implement a variant version that concatenates its initially generated answer with the instruction as condition to re-extract features. The initial generated answer serves as an additional heuristic from the LLM for guiding feature extraction. Then, the newly extracted visual prompts are used to re-generate answers. For a fair comparison, we provide a zero-shot version (Row 5) and a fine-tuned version (Row 6) using synthetic training as ours. As shown in Table 4, directly using synthetic data and inferred answers as heuristic conditions fails to yield a notable improvement. In contrast, VPG-C can better guide the VPG to complete the missing visual details by intercepting the intermediate representations of the LLM. Further, VPG-C is more computation-efficient as it only requires one full forward pass of the LLM, while the InstructBLIP variants require twice.

**VPG-C works well on various language backbones.** Table 4 also validates that our approach can well cooperate with language backbones of *different families (LLaMA2) and sizes (Vicuna-13B).*

**VPG-C can be implemented with very simple VPG.** As a generic method, VPG-C can be implemented with different VPGs. Beyond the widely used Q-former that is composed of multiple Transformer blocks, we further probe the effectiveness of VPG-C with a simpler VPG, *i.e.,* Linear Projection, as used in LLaVA (please refer to Appendix C for implementation details). Table 4 Row 7 shows promising results. VPG-C can also significantly bolster the performance of a simple

Table 4: Ablation results on DEMON Benchmark.

| | | Multimodal Dialogue | Visual Storytelling | Visual Relation Inference | Multimodal Cloze | Knowledge Grounded QA | Text-Rich Images QA | Multi-Image Reasoning |
|---|---|---|---|---|---|---|---|---|
| 1 | Backbone | 25.65 | 21.72 | 9.33 | 17.06 | 37.21 | 32.42 | 41.30 |
| 2 | +Heuristic Details | 28.13 | 22.76 | 12.69 | 18.81 | 38.75 | 34.14 | 43.26 |
| 3 | +VPG-C | 31.76 | 23.62 | 19.12 | 20.09 | 42.53 | 39.68 | 46.71 |
| 4 | +Synthetic Training | 37.50 | 25.20 | 25.90 | 22.15 | 48.60 | 44.93 | 50.28 |
| | InstructBLIP | 33.58 | 24.41 | 11.48 | 21.20 | 47.40 | 44.40 | 48.55 |
| 5 | +Answer Condition | 32.10 | 23.76 | 11.02 | 21.86 | 47.94 | 42.08 | 49.01 |
| 6 | +Synthetic Training | 31.76 | 24.32 | 12.78 | 19.87 | 46.58 | 42.36 | 49.82 |
| | LLaVA | 7.79 | 10.70 | 8.27 | 15.85 | 36.20 | 28.33 | 41.53 |
| 7 | Linear VPG | 16.43 | 19.48 | 14.75 | 18.54 | 41.32 | 36.87 | 46.02 |
| 8 | VPG-C-LLaMA2-7B | 42.70 | 24.76 | 25.50 | 22.95 | 51.00 | 44.93 | 48.68 |
| 9 | VPG-C-Vicuna-13B | 38.14 | 26.59 | 27.15 | 27.15 | 52.93 | 49.33 | 53.65 |

linear VPG, verifying the transferability of VPG-C. It is promising to adapt our generic VPG-C and corresponding low-resource synthetic training strategy to different VPGs in the future.

**Analysis on the inserted layer of VPG-C.** We investigate the impact of inserting VPG-C into different layers of LLMs. We report the averaged accuracy for multiple-choice tasks and averaged ROUGE-L for open-ended generation tasks in Figure 6. We observe that the performance is low when we insert VPG-C too early (*i.e.,* 4, 8) as the model might not have gathered sufficient contextual information to generate effective guidance. Meanwhile, inserting VPG-C too late (*i.e.,* 24, 28) degenerates the performance. We speculate this is due to the generated guidance being too concentrated and there not being enough layers to integrate the residual details.

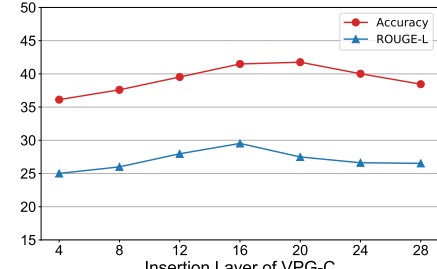

Figure 6: Performance on DEMON with different insertion layers.

**Synthetic training is data-efficient.** Since our proposed synthetic training strategy can construct challenging discriminative tasks in a targeted manner, enhancing VPG-C's ability to complete missing details, it avoids the need for a large amount of supervised demonstrative instruction data. We further investigate the impact of different numbers of synthetic training data. As illustrated in Table 5, the performance keeps increasing when the number of data is increased from 16K to 64K. Beyond this, escalating the data count from 64K

Table 5: Efficiency analysis of synthetic training.

| | Accuracy | ROUGE-L |
|---|---|---|
| 16K | 38.93 | 25.67 |
| 32K | 39.62 | 27.38 |
| 48K | 40.45 | 28.81 |
| 64K | **41.49** | **29.53** |
| 80K | 41.62 | 29.73 |
| 96K | 40.12 | 28.31 |

to 80K yields only marginal enhancement. Further amplification of data eventually triggers a performance dip as excessive data leads to model overfitting to the synthetic training task.

**Image order sensitivity.** The order of interleaved images in demonstrative instructions is pivotal for the compositional semantics of the instruction. Intuitively, altering the order of images within a demonstrative instruction can significantly shift its semantics. Consequently, variations in model performance can reveal the model's sensitivity to the instruction semantics. An ideal model should keenly capture changes in instruction semantics. Therefore, we visualize the performance variations of models by randomly shuffling the order of interleaved images within the demonstrative instructions. According to Figure 7, we surprisingly find that SOTA models are less sensitive to the image order. In contrast, VPG-C can keenly capture the semantic changes caused by the shuffled image order. Particularly, our performance varies dramatically in multimodal dialogue, as the order of images within these tasks is closely intertwined with the dialogue content.

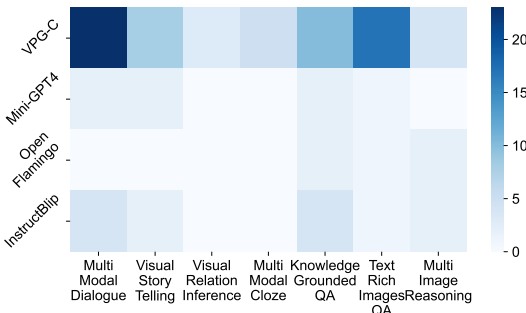

Figure 7: Analysis on image order sensitivity.

**Qualitative examples.** As illustrated in Figure 8, VPG-C demonstrates strong abilities to perform reasoning over complicated demonstrative instructions. For instance, in **(a)**, VPG-C can keenly identify the connections between the images and thereby infer the reason that causes this unusual phenomenon. In **(b, c)**, VPG-C exhibits the ability to comprehend absurd objects through multimodal conversations with humans. In **(d, e)**, VPG-C can reasonably infer the relations among

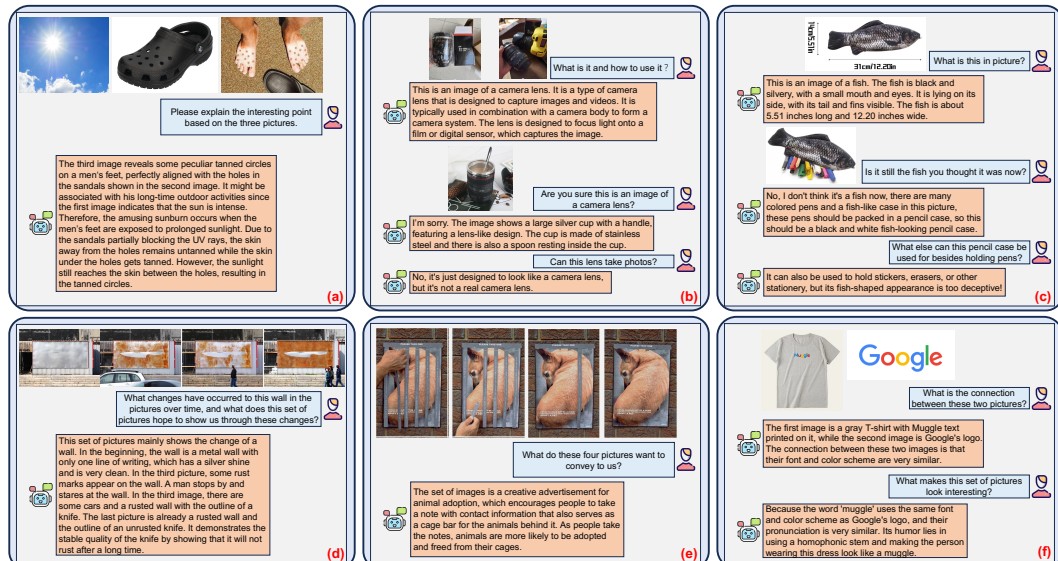

Figure 8: Qualitative examples generated by our `VPG-C-Vicuna-7B` model.

the images and understand the metaphorical implications they want to convey. In Appendix G, we provide more practical examples as well as comparisons with other MLLMs, where we find that baseline models fail to correctly associate multiple images and comprehend demonstrative context.

## 5 RELATED WORK

MLLMs (Yin et al., 2023) aim to serve as a general-purpose assistant to perform various vision-language tasks by free-text generation. Flamingo (Alayrac et al., 2022) and BLIP-2 (Li et al., 2023c) bridge LLMs with powerful pre-trained visual encoders and demonstrate strong zero-shot ability by aligning visual features with LLMs. Follow-up works of LLaVA (Liu et al., 2023), MiniGPT-4 (Zhu et al., 2023a), InstructBLIP (Dai et al., 2023), Hallucidoctor (Yu et al., 2024), mPLUG-Owl (Ye et al., 2023) propose to fine-tune MLLMs with multimodal instruction tuning data. To effectively benchmark the recent progress in MLLMs, concurrent works of LVLM-eHub (Xu et al., 2023) and MME Benchmark (Fu et al., 2023) are proposed, while they mainly focus on instructions that only involve a single image with limited instruction diversity. In this paper, we propose the first demonstrative instruction-following benchmark, covering various tasks of diverse scenarios. Further, we propose a lightweight and generic `VPG-C` module to address the inherent limitation of current VPGs. Our `VPG-C` is efficiently tuned by our synthetic discriminative training strategy, which demonstrates powerful potentials of text-to-image diffusion models (He et al., 2022; Lin et al., 2023; Prabhu et al., 2023; Bansal & Grover, 2023; Yu et al., 2023b) to facilitate vision-language understanding (Radford et al., 2021b; Jia et al., 2021; Li et al., 2022b).

## 6 CONCLUSION

In this paper, we propose `VPG-C`, a generic and parameter-efficient approach that infers and completes the missing visual details for MLLMs to comprehend demonstrative instructions with interleaved multimodal context. Meanwhile, we present a synthetic discriminative training strategy to fine-tune `VPG-C`, eliminating the need for supervised demonstrative instruction data. To foster the research on demonstrative instruction understanding, we build `DEMON`, a comprehensive benchmark for multimodal large language models, consisting of 31 tasks with complicated vision-language demonstrative context, covering a wide range of scenarios. Through synthetic training, `VPG-C` showcases notable zero-shot performance on the `DEMON` benchmark. Its superior performance on other established benchmarks like MME and OwlEval further underscores its effectiveness.

**Acknowledgment.** This work was supported by the NSFC (No. 62272411), Key Research and Development Projects in Zhejiang Province (No. 2024C01106), the National Key Research and Development Project of China (2018AAA0101900), the Tencent WeChat Rhino-Bird Special Research Program (Tencent WXG-FR-2023-10), and Research funding from FinVolution Group.

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

# A OVERVIEW

In this appendix, we present:

- Detailed information of the proposed DEMON benchmark (Section B).
- Implementation details of our VPG-C (Section C).
- Implementation details of existing MLLMs on the DEMON benchmark (Section D).
- Detailed zero-shot performance on MME benchmark (Section E).
- Detailed zero-shot performance on DEMON benchmark (Section F).
- Qualitative comparison with existing MLLMs (Section G).

# B BENCHMARK DETAILS

| Task | Scenario | Dataset | Metirc |
|---|---|---|---|
| **Multimodal Dialogue** | | | |
| Conversational Embodied Dialogue | Embodied | ALFRED (Shridhar et al., 2020) | ROUGE-L |
| Multimodal Dialogue | Conversation | MMCoQA (Li et al., 2022e) | ROUGE-L |
| **Visual Storytelling** | | | |
| Animated Story Completion | Cartoon | AESOP (Ravi et al., 2021) | ROUGE-L |
| Animated Story Completion | Cartoon | PororoSV (Li et al., 2019) | ROUGE-L |
| Animated Story Completion | Cartoon | FlintstonesSV (Gupta et al., 2018) | ROUGE-L |
| Sequential Photo Storytelling | Album | VIST (Huang et al., 2016) | ROUGE-L |
| Sequential Photo Storytelling | Cartoon | DiDeMoSV (Maharana et al., 2022) | ROUGE-L |
| **Visual Relation Inference** | | | |
| Visual Change Captioning | Surveillance | Spot-the-Diff (Jhamtani & Berg-Kirkpatrick, 2018) | ROUGE-L |
| Visual Change Captioning | Synthetic | CLEVR-Change (Hosseinzadeh & Wang, 2021) | ROUGE-L |
| Visual Relationship Expressing | General | IEdit (Tan et al., 2019) | ROUGE-L |
| Subtle Difference Expressing | Fine-Grained | Birds-to-Words (Forbes et al., 2019) | ROUGE-L |
| **Multimodal Cloze** | | | |
| Comic Dialogue Identification | Cartoon | COMICS-Dialogue (Iyyer et al., 2017) | Accuracy |
| Comic Panel Identification | Cartoon | COMICS-Panel (Iyyer et al., 2017) | Accuracy |
| Recipe Completion | Recipe | RecipeQA-TextCloze (Yagcioglu et al., 2018) | Accuracy |
| Visual Step Cloze | Recipe | RecipeQA-VisualCloze (Yagcioglu et al., 2018) | Accuracy |
| **Knowledge Grounded QA** | | | |
| Webpage QA | Webpage | WebQA (Chang et al., 2022) | Accuracy |
| Textbook QA | Textbook | TQA (Kembhavi et al., 2017) | Accuracy |
| Complex Multimodal QA | Wikipedia | MMQA (Talmor et al., 2021) | Accuracy |
| Complex Multimodal QA* | Wikipedia | MANYMODALQA (Hannan et al., 2020) | Accuracy |
| **Text-Rich Images QA** | | | |
| Slide QA | Slide | SlideVQA (Tanaka et al., 2023) | Accuracy |
| OCR QA | Book Cover | OCR-VQA (Mishra et al., 2019) | Accuracy |
| Document QA | Document Image | DocVQA (Mathew et al., 2021) | Accuracy |
| **Multi-Image Reasoning** | | | |
| Image-Set QA* | Indoor Egocentric | Gibson (Bansal et al., 2020; Xia et al., 2018) | Accuracy |
| Image-Set QA | Driving Recording | nuScenes (Bansal et al., 2020; Caesar et al., 2020) | Accuracy |
| Industrial Inspection | Industrial | VISION (Bai et al., 2023) | Accuracy |
| Fashion QA | Fashion | Fashion200K (Han et al., 2017) | Accuracy |
| Property Coherence | General | MIT-States-PropertyCoherence (Isola et al., 2015) | Accuracy |
| State Transformation Coherence | General | MIT-States-StateCoherence (Isola et al., 2015) | Accuracy |
| Visual Step Matching | Recipe | RecipeQA-ImageCoherence (Yagcioglu et al., 2018) | Accuracy |
| Multi-Image Visual Entailment | General | NLVR2 (Suhr et al., 2018) | Accuracy |
| Ambiguity Analysis | Mobile Photo | VizWiz (Bhattacharya et al., 2019) | Accuracy |

Table 6: Summary of the demonstrative instruction-following tasks in DEMON benchmark. * indicates the tasks that are not included in DEMON-Core.

# C IMPLEMENTATION DETAILS

**Model.** We choose ViT-G/14 from EVA-CLIP (Fang et al., 2023) as our visual encoder and pretrained Q-former from BLIP-2 without instruction tuning as the task-agnostic visual prompt generator. For the large language model, we implement three versions: LLaMA2-7B (Touvron et al.,

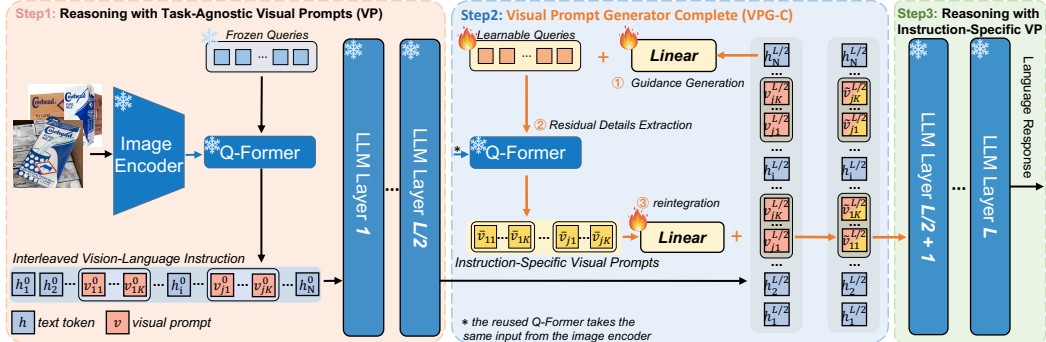

Figure 9: Detailed framework of our MLLM enhanced with `VPG-C`.

2023b), Vicuna-7B (Chiang et al., 2023), Vicuna-13B, with 32, 32, 48 Transformer layers, respectively. We derive instruction-specific conditions from the 16th / 24th layer and re-inject the conditional visual knowledge into the 17th / 25th layer. Furthermore, we provide detailed framework of our MLLM enhanced with `VPG-C` in Figure 9.

**Choice of Q-former.** Recently, InstructBLIP (Dai et al., 2023) proposes to take the instruction as additional input to the Q-former and fine-tune the Q-former to extract visual features according to instructions using 16M multimodal instruction tuning data. While achieving outstanding performance on in-domain tasks, a recent study (Xu et al., 2023) indicates that fine-tuning on massive in-domain data severely undermines its generalizability on open-world scenarios. Instead of directly relying on the Q-former to achieve task-specific feature extraction by massive instruction tuning, we aim to utilize the sophisticated reasoning ability of LLMs to guide the Q-former to conditionally attend to residual visual details. Thus, we use the Q-former without instruction data tuning from BLIP-2 (Li et al., 2023c), which extracts the task-agnostic primary visual contents at the first time.

**Training.** We implement `VPG-C` in LAVIS library (Li et al., 2022a). We keep the visual backbone, visual prompt generator, and the language model frozen, and tune the `VPG-C` module using the proposed training strategy. Since BLIP-2 models do not include pre-trained Q-former that matches Vicuna and LLaMA2, we reuse the Q-former that matches FlanT5-XXL and fine-tune the last linear projection layer with 5 million image-text pairs to align it with Vicuna/LLaMA2. All the tunable parameters of our `VPG-C` module are a set of query embeddings and two linear projection layers, which only accounts for 0.09% ($\sim$6.3M) of the entire model. As for synthetic training, we select about 30k images from CC3M (Sharma et al., 2018) that contain significantly ignored objects and perform different types of editing on them. Totally, we generate approximately 64k synthetic images with suitable modifications. To stablize the training and avoid overfitting, we use 500k image-caption pairs from CC3M to jointly train the `VPG-C` module. We tune the `VPG-C` module for 18k steps using a batch size of 24 for synthetic training and 64 for image captioning, which takes about 7 hours to complete with a single A100 GPU. Additionally, we adopt the AdamW optimizer with $\beta = (0.9, 0.999)$, and set the learning rate and weight decay to 0.00002 and 0.05, respectively. We warm up the training with 2k warm-up steps, followed by a learning rate decay mechanism with the cosine schedule.

**Implementation of `VPG-C` with the linear VPG.** As a generic method, `VPG-C` can be implemented with different VPGs. Beyond widely used Q-former that is composed of multiple Transformer blocks, we further probe the effectiveness of `VPG-C` with a simpler VPG, i.e., Linear Projection, as used in LLaVA (Liu et al., 2023). LLaVA trains a simple linear layer as the VPG to connect image features into the word embedding space. To implement `VPG-C` with the linear VPG, we first linearly project the generated guidance $\mathbf{g}$ and then take it as a filter to perform element-wise Hadamard product with the visual features $\mathcal{X}^I$ from the image encoder:

$$\overline{\mathcal{V}} = (\mathcal{W}_1 \mathbf{g} \mathbf{1}^T) \odot (\mathcal{W}_2 \mathcal{X}^I) \tag{1}$$

where $\mathcal{W}_1$ and $\mathcal{W}_2$ are linear projection matrixes, $\mathbf{1}^T$ is the transpose of an all-ones vector, and $\odot$ represents Hadamard product. The output $\overline{\mathcal{V}}$ represents the newly-extracted missing visual details according to the inferred guidance. And $\overline{\mathcal{V}}$ is reintegrated into the LLM in the same manner.

## D    MODEL DETAILS IN DEMON BENCHMARK

Recent advancements in LLMs (OpenAI, 2023a;b) have heralded significant achievements across various domains. Inspired by this success, many MLLMs (Li et al., 2023c; Liu et al., 2023; Zhu et al., 2023a; Alayrac et al., 2022; Ye et al., 2023; Gao et al., 2023; Li et al., 2023a) have been proposed to foster generalist vision-language reasoning (Xu et al., 2015; Li et al., 2023b; 2020; Yu et al., 2023a; Li et al., 2022d; Zhang et al., 2022; Li et al., 2022c; 2023d; Zhang et al., 2019; Antol et al., 2015). In our experiments, we conducted comparisons with some of the most recent and representative MLLMs in the following.

- **LLaVA** (Liu et al., 2023) establishes a connection between the visual encoder ViT-L/14 from CLIP (Radford et al., 2021a) and the language decoder LLaMA (Touvron et al., 2023a), utilizing a lightweight, fully-connected (FC) layer. Initially, the system trains this FC layer using 595K image-text pairs, while keeping both the visual encoder and LLM static. Following this, LLaVA fine-tunes both the FC layer and LLM using a dataset comprising 158K instructional vision-language pairs. The tested version is "LLaVA-7B-v0".

- **LLaMA-Adapter V2** (Gao et al., 2023) stands as a model of parameter efficiency within the realm of visual instruction. Despite maintaining the visual encoder (ViT-L/14) and the LLM in a static state, LA-V2 distributes the instruction-following capacity of the entire LLaMA system via bias-tuning. This method allows for the refinement of scale, bias, norm, and prompt parameters on diverse data sets. These include 200M image captioning data, 158K visual instruction-following data, and an additional 52K language instruction-following data, the latter of which was assembled by GPT-4 (OpenAI, 2023b). The tested version is "LLaVA-7B".

- **MiniGPT-4** (Zhu et al., 2023a) bridges the gap between the visual encoder and text encoder using a fully-connected (FC) layer. Initially, this model trains the FC layer on a dataset comprised of 5M image-text pairs before fine-tuning it on 3.5K instructional vision-language data. Notwithstanding its simplicity, MiniGPT-4 requires the loading of a pre-trained vision encoder from BLIP2, as well as a Vicuna LLM (Chiang et al., 2023). The tested version is "minigpt4-aligned-with-vicuna7b".

- **BLIP2** (Li et al., 2023c) employs a dual-stage strategy to seamlessly bridge the modality gap, utilizing a lean Q-Former pre-trained on 129 million image-text pairs. The initial stage kick-starts the learning process of vision-language representation, leveraging a frozen image encoder, the ViT-g/14 from EVA-CLIP (Fang et al., 2023). Subsequently, the second stage harnesses a frozen LLM, the FlanT5 (Chung et al., 2022), to initiate the vision-to-language generative learning. This innovative strategy effectively facilitates zero-shot instructed image-to-text generation. The tested version is "blip2-pretrain-flant5xl".

- **mPLUG-Owl** (Ye et al., 2023) introduces a visual abstractor, fundamentally close the Perceiver Resampler in Flamingo (Alayrac et al., 2022), as a bridge between the pre-trained visual encoder ViT-L/14 and the LLM (LLaMA (Touvron et al., 2023a)). This model adopts a two-stage fine-tuning procedure. In the initial phase, both the visual encoder and the visual abstractor undergo comprehensive fine-tuning using a dataset of 204M image-text pairs. Subsequently, in the second phase, mPLUG-Owl applies the 158K LLaVA-Instruct dataset to fine-tune the pre-trained LLM in a parameter-efficient manner through the use of LoRA (Hu et al., 2021). The tested version is "mplug-owl-llama-7b".

- **Otter** (Li et al., 2023a) is a multimodal model that applies in-context instruction tuning based on OpenFlamingo (Alayrac et al., 2022). This model integrates a LLaMA-7B (Touvron et al., 2023a) language encoder and a CLIP ViT-L/14. While the visual and text encoders remain static, Otter refines an additional 1.3 billion parameters. These parameters are derived from adaptation modules and are trained using 158K instruction-following data. The tested version is "OTTER-Image-LLaMA7B-LA-InContext".

- **InstructBLIP** (Dai et al., 2023) originates from a pre-trained BLIP-2 model, which consists of a ViT-g/14 image encoder, a Vicuna, and a Q-Former to act as the bridge between these two components. During the process of vision-language instruction tuning, only the Q-Former undergoes fine-tuning, with the training process leveraging data from 13 distinct visual question-answering datasets. The tested version is "blip2-vicuna-instruct-7b".

- **OpenFlamingo** (Alayrac et al., 2022; Awadalla et al., 2023) represents one of the pioneering efforts to incorporate Language Model Learning (LLMs) into the domain of vision-language pretraining. To optimize its conditioning on visual features, Flamingo strategically integrates a number of gated cross-attention dense blocks amidst the layers of the pre-trained language encoder. OpenFlamingo offers an open-source rendition of this advanced model. The tested version is "llama-7b".

The `DEMON` benchmark predominantly features interleaved vision-language instructions, distinguishing it from the traditional single-image datasets. While our innovative method, `VPG-C`, along with OpenFlamingo and MiniGPT-4, inherently accommodates interleaved image-text sequences, other models like BLIP-2, InstructBlip, LLaVA, mPLUG-Owl, Otter, and LLaMA-Adapter V2 do not. For these, we employed a strategy where we concatenate the embeddings of all images. This approach can be analogized to treating images as frames within a video. To maintain the positional context of each image in an interleaved image-text instruction, we explicitly indicate the location of each image within the context.

## E   DETAILED ZERO-SHOT PERFORMANCE ON MME BENCHMARK

In this section, we report the detailed performance on the 14 subtasks of MME benchmark in Table 7.

Table 7: Detailed zero-shot performance on MME benchmark.

|  | BLIP-2 | InstructBLIP | LA-V2 | LLaVA | MiniGPT-4 | mPLUG-Owl | Otter | VPG-C |
|---|---|---|---|---|---|---|---|---|
| Existence | 160.00 | 185.00 | 120.00 | 50.00 | 115.00 | 120.00 | 195.00 | 180.00 |
| Count | 135.00 | 143.33 | 50.00 | 50.00 | 123.33 | 88.33 | 50.00 | 96.67 |
| Position | 73.33 | 66.67 | 48.33 | 50.00 | 81.67 | 50.00 | 86.67 | 80.00 |
| Color | 148.33 | 153.33 | 75.00 | 55.00 | 110.00 | 55.00 | 113.33 | 116.67 |
| Poster | 141.84 | 123.81 | 99.66 | 50.00 | 55.78 | 136.05 | 138.78 | 147.28 |
| Celebrity | 105.59 | 101.18 | 86.18 | 48.82 | 65.29 | 100.29 | 172.65 | 164.12 |
| Scene | 145.25 | 153.00 | 148.50 | 50.00 | 95.75 | 135.50 | 158.75 | 156.00 |
| Landmark | 138.00 | 79.75 | 150.25 | 50.00 | 69.00 | 159.25 | 137.25 | 145.00 |
| Artwork | 136.50 | 134.25 | 69.75 | 49.00 | 55.75 | 96.25 | 129.00 | 113.50 |
| OCR | 110.00 | 72.50 | 125.00 | 50.00 | 95.00 | 65.00 | 72.50 | 100.00 |
| Perception | 1293.84 | 1212.82 | 972.67 | 502.82 | 866.57 | 967.34 | 1292.26 | 1299.24 |
| Commonsense | 110.00 | 129.29 | 81.43 | 57.14 | 72.14 | 78.57 | 106.43 | 98.57 |
| Numerical | 40.00 | 40.00 | 62.50 | 50.00 | 55.00 | 60.00 | 72.50 | 77.50 |
| Text Translation | 65.00 | 65.00 | 50.00 | 57.50 | 55.00 | 80.00 | 57.50 | 57.50 |
| Code Reasoning | 75.00 | 57.50 | 55.00 | 50.00 | 110.00 | 57.50 | 70.00 | 87.50 |
| Cognition | 290.00 | 291.79 | 248.93 | 214.64 | 292.14 | 276.07 | 306.43 | 321.07 |

## F   DETAILED ZERO-SHOT PERFORMANCE ON DEMON BENCHMARK

Table 8: Zero-shot evaluation on multimodal dialogue.

|  | Conversational Embodied Dialogue | Multimodal Dialogue |
|---|---|---|
| BLIP-2 | 16.75 | 35.49 |
| InstructBLIP | 18.07 | 49.09 |
| LLaMA-Adapter V2 | 19.04 | 9.40 |
| LLaVA | 10.19 | 5.39 |
| MiniGPT-4 | 16.82 | 10.57 |
| mPLUG-Owl | 11.07 | 14.27 |
| OpenFlamingo | 24.27 | 9.49 |
| Otter | 16.06 | 14.68 |
| VPG-C-LLaMA2-7B | 48.31 | 37.04 |
| VPG-C-Vicuna-7B | 41.02 | 33.99 |
| VPG-C-Vicuna-13B | 42.25 | 34.02 |

Table 9: Zero-shot evaluation on visual storytelling.

| | Animated Story Completion-AESOP | Animated Story Completion-PororoSV | Animated Story Completion-FlintstonesSV | Sequential Photo Storytelling-VIST | Sequential Photo Storytelling-DiDeMoSV |
|---|---|---|---|---|---|
| BLIP-2 | 21.64 | 26.24 | 29.61 | 13.16 | 24.2 |
| InstructBLIP | 18.80 | 28.20 | 33.32 | 16.92 | 24.80 |
| LLaMA-Adapter V2 | 18.01 | 20.15 | 24.22 | 10.89 | 14.57 |
| LLaVA | 13.56 | 11.44 | 12.77 | 8.00 | 7.71 |
| MiniGPT-4 | 12.23 | 16.00 | 26.48 | 14.82 | 15.81 |
| mPLUG-Owl | 18.28 | 20.49 | 32.12 | 10.82 | 14.94 |
| OpenFlamingo | 23.32 | 32.35 | 37.79 | 15.14 | 12.50 |
| Otter | 13.94 | 17.52 | 22.21 | 9.96 | 14.23 |
| VPG-C-LLaMA2-7B | 19.98 | 28.67 | 38.14 | 16.95 | 20.05 |
| VPG-C-Vicuna-7B | 19.93 | 28.36 | 39.19 | 17.34 | 21.27 |
| VPG-C-Vicuna-13B | 20.53 | 29.81 | 41.32 | 19.04 | 22.26 |

Table 10: Zero-shot evaluation on visual relation inference.

| | Visual Change Captioning -Spot-the-Diff | Visual Change Captioning -CLEVR-Change | Visual Relationship Expressing | Subtle Difference Expressing |
|---|---|---|---|---|
| BLIP-2 | 17.48 | 3.21 | 12.37 | 9.62 |
| InstructBLIP | 19.71 | 4.61 | 10.70 | 10.92 |
| LLaMA-Adapter V2 | 16.72 | 15.52 | 7.88 | 13.92 |
| LLaVA | 8.50 | 8.76 | 6.72 | 9.11 |
| MiniGPT-4 | 7.50 | 7.49 | 7.84 | 8.97 |
| mPLUG-Owl | 6.06 | 1.46 | 6.22 | 7.86 |
| OpenFlamingo | 13.01 | 11.90 | 12.57 | 17.90 |
| Otter | 12.69 | 11.63 | 8.85 | 12.38 |
| VPG-C-LLaMA2-7B | 21.02 | 42.05 | 14.10 | 24.81 |
| VPG-C-Vicuna-7B | 20.01 | 41.60 | 16.35 | 25.64 |
| VPG-C-Vicuna-13B | 21.56 | 40.67 | 20.27 | 26.08 |

Table 11: Zero-shot evaluation on multimodal cloze.

| | Comic Dialogue Identification | Comic Panel Identification[1] | Recipe Completion | Visual Step Cloze[1] |
|---|---|---|---|---|
| BLIP-2 | 39.70 | 0.00 | 30.46 | 1.60 |
| InstructBLIP | 40.60 | 0.00 | 27.40 | 16.80 |
| LLaMA-Adapter V2 | 24.40 | 0.40 | 38.20 | 9.00 |
| LLaVA | 30.60 | 0.00 | 32.80 | 0.00 |
| MiniGPT-4 | 33.00 | 1.00 | 31.60 | 0.80 |
| mPLUG-Owl | 36.60 | 0.00 | 27.60 | 0.80 |
| OpenFlamingo | 38.40 | 1.20 | 29.00 | 18.00 |
| Otter | 29.00 | 0.00 | 35.00 | 0.00 |
| VPG-C-LLaMA2-7B | 36.80 | 1.80 | 51.80 | 1.40 |
| VPG-C-Vicuna-7B | 39.20 | 3.60 | 30.40 | 15.40 |
| VPG-C-Vicuna-13B | 42.20 | 8.20 | 39.80 | 18.40 |

[1] For tasks with images as options, only responses that begin with the correct answer will be evaluated as correct.

Table 12: Zero-shot evaluation on knowledge grounded QA.

| | Webpage QA | Textbook QA | Complex Multimodal QA |
|---|---|---|---|
| BLIP-2 | 47.60 | 29.73 | 40.36 |
| InstructBLIP | 45.20 | 30.20 | 66.80 |
| LLaMA-Adapter V2 | 44.60 | 46.00 | 43.80 |
| LLaVA | 39.40 | 39.60 | 29.60 |
| MiniGPT-4 | 27.40 | 28.60 | 34.80 |
| mPLUG-Owl | 34.20 | 30.00 | 35.60 |
| OpenFlamingo | 37.80 | 32.40 | 25.80 |
| Otter | 45.00 | 39.00 | 41.00 |
| VPG-C-LLaMA2-7B | 49.40 | 42.40 | 61.20 |
| VPG-C-Vicuna-7B | 50.00 | 33.40 | 62.40 |
| VPG-C-Vicuna-13B | 50.60 | 43.40 | 64.80 |

Table 13: Zero-shot evaluation on text-rich images QA.

|  | Slide QA | OCR QA | Document QA |
|---|---|---|---|
| BLIP-2 | 43.80 | 10.40 | 46.40 |
| InstructBLIP | 42.00 | 44.20 | 47.00 |
| LLaMA-Adapter V2 | 43.00 | 3.40 | 49.60 |
| LLaVA | 38.80 | 2.60 | 43.60 |
| MiniGPT-4 | 35.20 | 7.20 | 36.80 |
| mPLUG-Owl | 35.60 | 22.60 | 39.20 |
| OpenFlamingo | 35.60 | 3.80 | 52.40 |
| Otter | 38.40 | 2.20 | 42.60 |
| VPG-C-LLaMA2-7B | 45.80 | 39.60 | 49.40 |
| VPG-C-Vicuna-7B | 46.80 | 39.40 | 48.60 |
| VPG-C-Vicuna-13B | 48.80 | 46.60 | 52.60 |

Table 14: Zero-shot evaluation on multi-image reasoning.

|  | Image-Set QA | Industrial Inspection | Fashion QA | Property Coherence | State Transformation Coherence | Visual Step Matching [1] | Multi-Image Visual Entailment | Ambiguity Analysis |
|---|---|---|---|---|---|---|---|---|
| BLIP-2 | 34.60 | 42.80 | 43.20 | 59.00 | 38.20 | 0.20 | 53.40 | 45.80 |
| instructblip7b | 65.00 | 50.60 | 44.40 | 59.20 | 59.40 | 11.60 | 55.20 | 43.00 |
| LLaMA-Adapter V2 | 41.60 | 55.00 | 45.60 | 48.80 | 63.00 | 0.00 | 54.80 | 43.40 |
| LLaVA | 29.60 | 53.00 | 45.20 | 50.40 | 59.20 | 0.80 | 50.80 | 43.20 |
| MiniGPT-4 | 30.40 | 59.80 | 49.20 | 52.00 | 57.80 | 0.20 | 50.60 | 48.00 |
| mPLUG-Owl | 29.20 | 54.20 | 45.80 | 50.00 | 60.60 | 0.00 | 55.00 | 45.20 |
| OpenFlamingo | 25.80 | 52.20 | 44.20 | 59.60 | 51.40 | 2.20 | 53.60 | 44.00 |
| Otter | 44.80 | 69.80 | 47.00 | 51.40 | 46.40 | 0.00 | 49.00 | 42.40 |
| VPG-C-LLaMA2-7B | 62.60 | 61.40 | 46.00 | 56.60 | 57.80 | 0.00 | 53.80 | 51.20 |
| VPG-C-Vicuna-7B | 67.20 | 48.80 | 50.00 | 60.80 | 60.00 | 0.20 | 57.80 | 57.40 |
| VPG-C-Vicuna-13B | 73.40 | 54.00 | 51.00 | 63.20 | 63.40 | 2.60 | 60.20 | 61.40 |

[1] For tasks with images as options, only responses that begin with the correct answer will be evaluated as correct.

# G QUALITATIVE COMPARISON

In this section, we compare our model with existing MLLMs on some complicated demonstrative instructions.

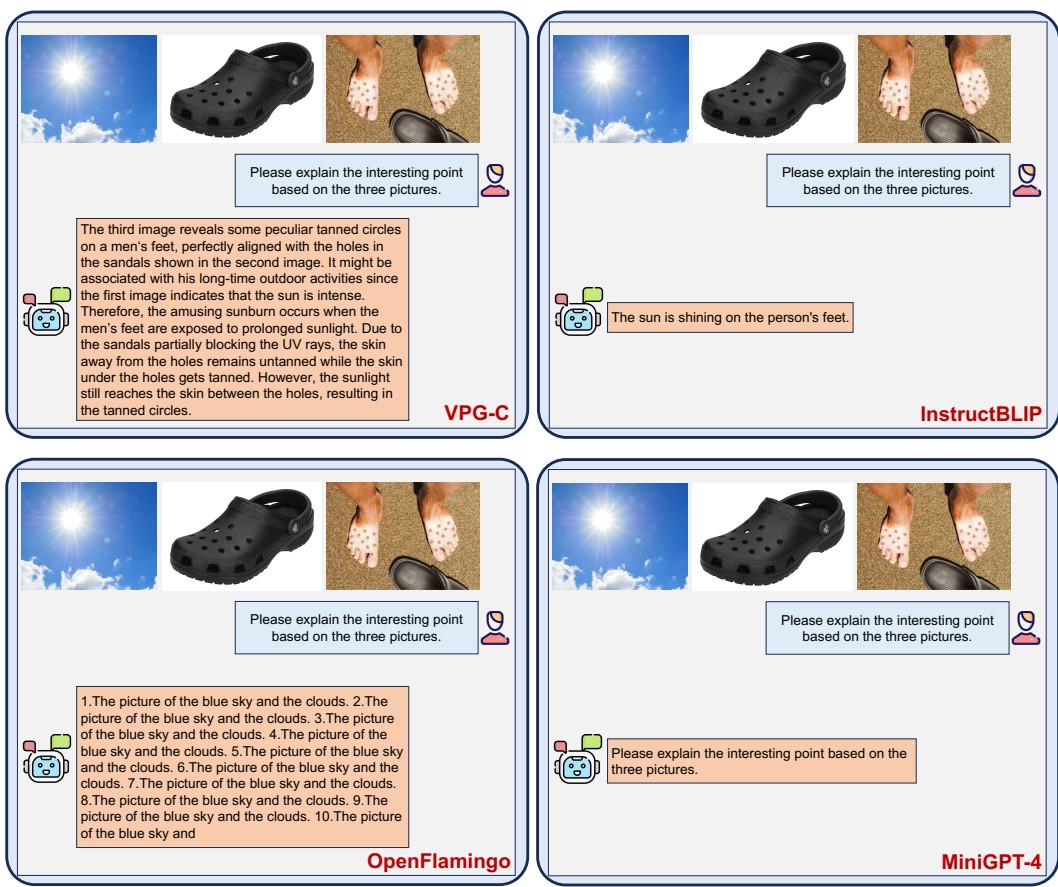

Figure 10: Qualitative comparison between our VPG-C and existing MLLMs.

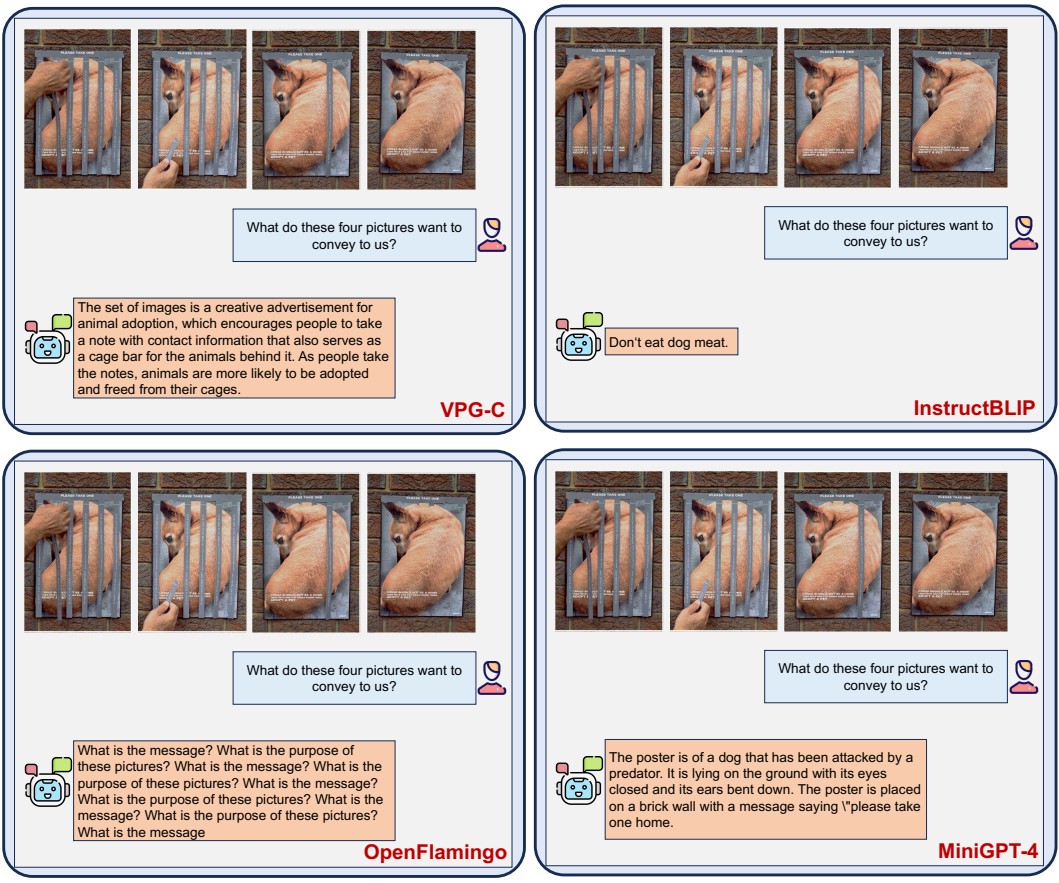

Figure 11: Qualitative comparison between our VPG-C and existing MLLMs.

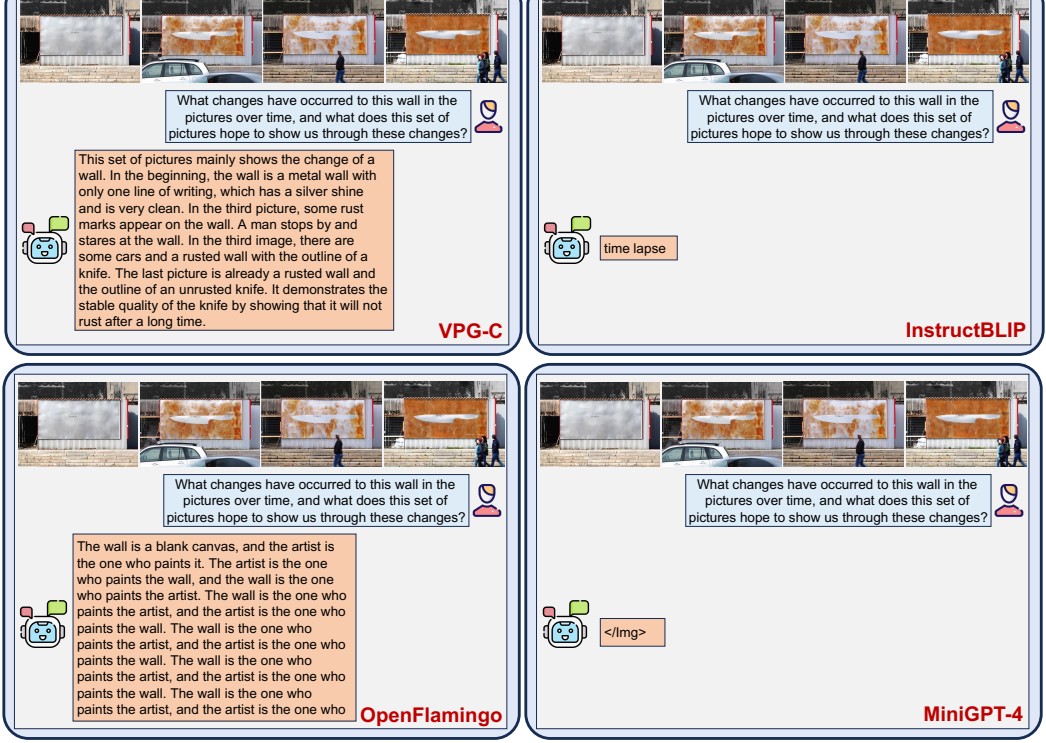

Figure 12: Qualitative comparison between our VPG-C and existing MLLMs.

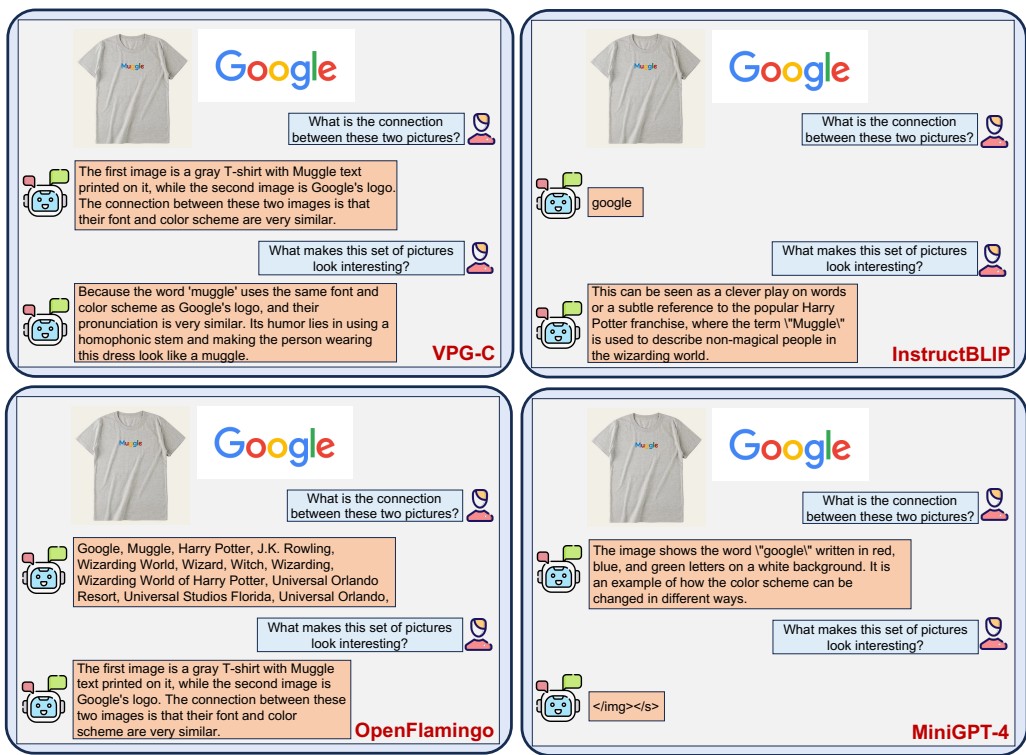

Figure 13: Qualitative comparison between our `VPG-C` and existing MLLMs.

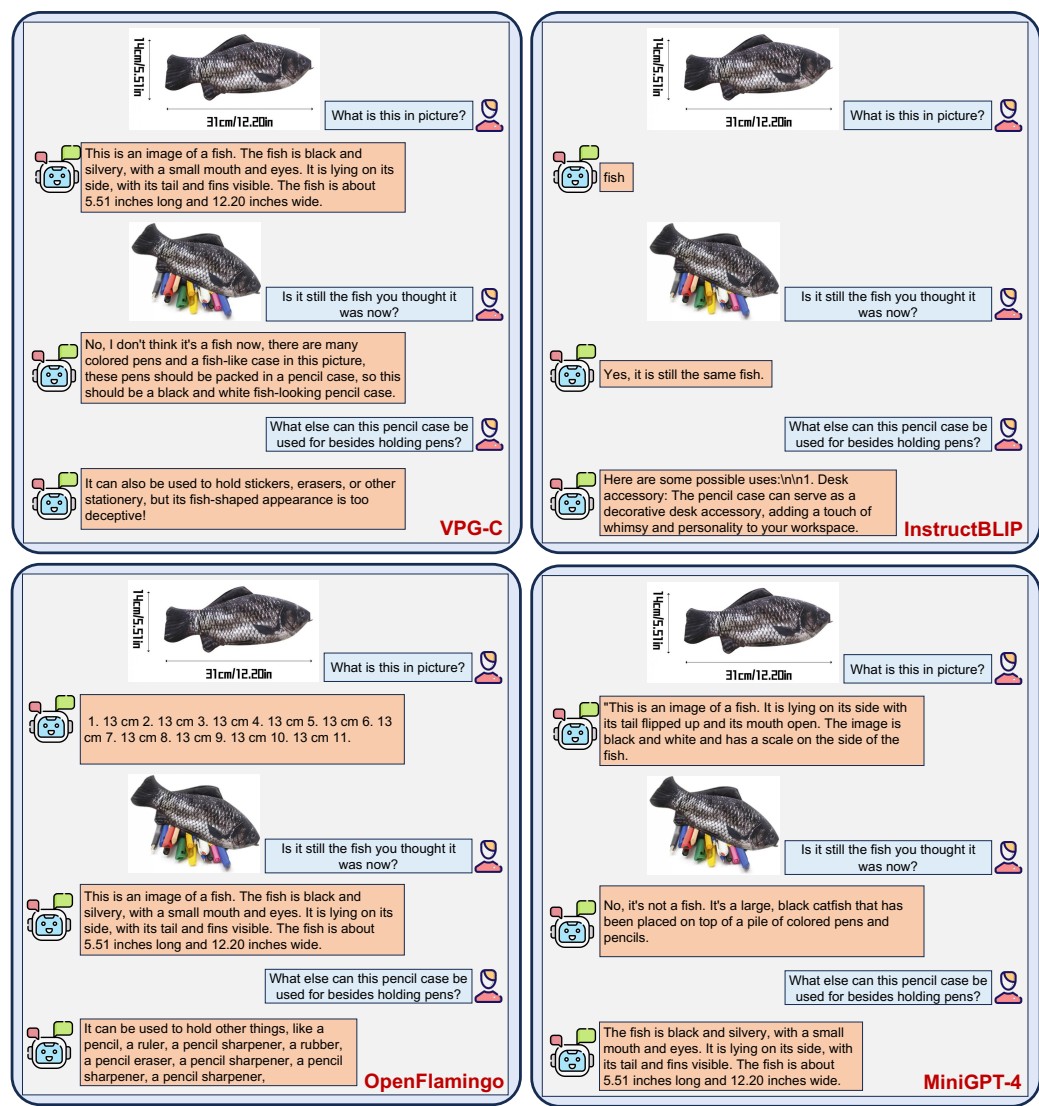

Figure 14: Qualitative comparison between our VPG-C and existing MLLMs.

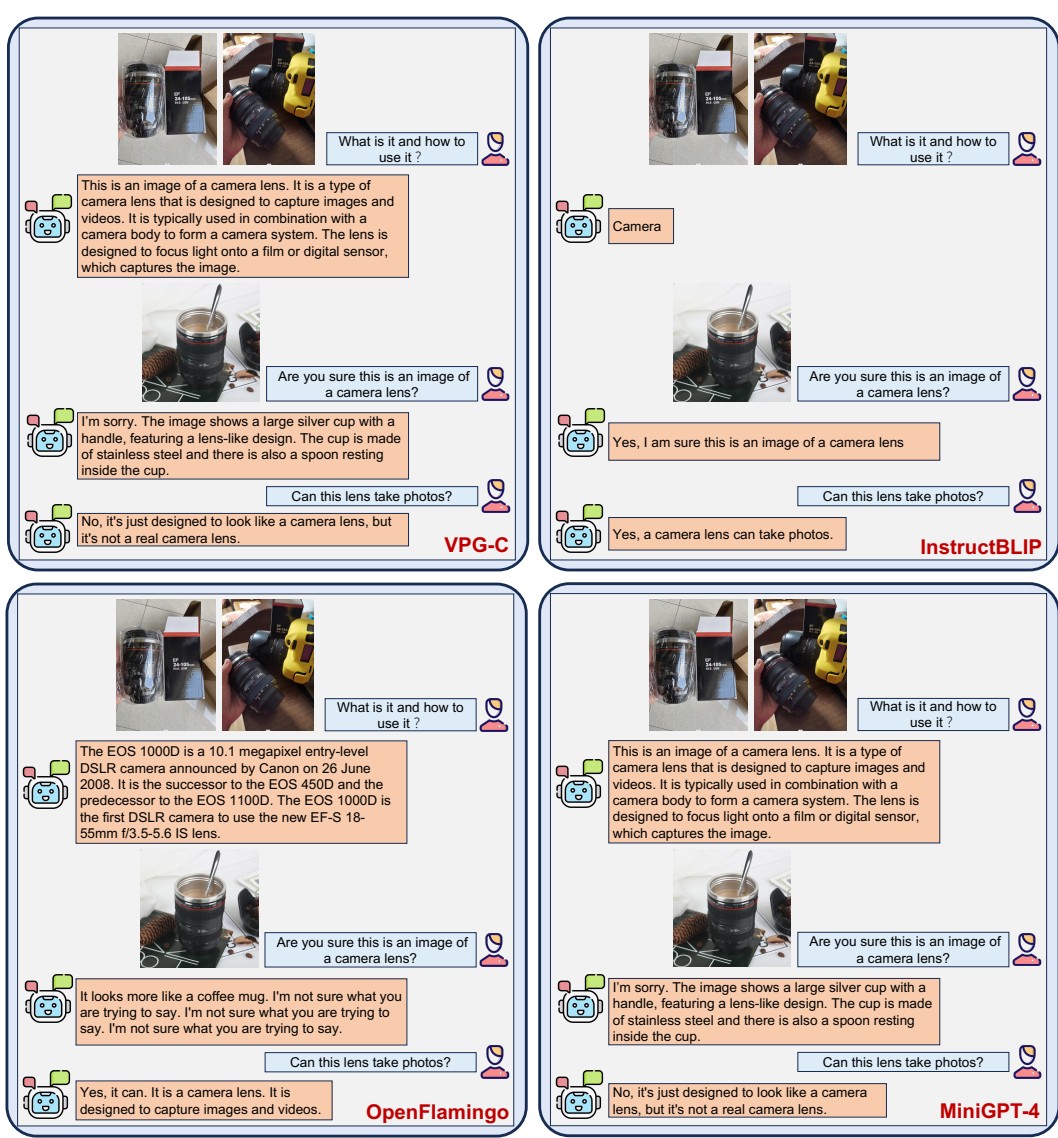

Figure 15: Qualitative comparison between our VPG-C and existing MLLMs.

