# OpenReview forum: "Fine-tuning Multimodal LLMs to Follow Zero-shot Demonstrative Instructions"
_ICLR.cc/2024/Conference — ICLR 2024 spotlight_

### Official Review · Reviewer_kFPK · 2023-10-30

**Soundness:** 2 fair
**Presentation:** 3 good
**Contribution:** 2 fair
**Rating:** 6
**Confidence:** 4

**Summary:**

This paper aims to address the issue that the image-captioning based training objective often leads the visual prompt generators (VPGs) to neglect visual details. It proposes a VPG-C(omplete) module to complete the missing details and a synthetic discriminative training strategy to train VPG-C without the need for supervised instructions. The experiments are conducted on the proposed DEMON benchmark, the MME, and OwlEval benchmarks.

**Strengths:**

- The paper is well-written and easy to follow.
- The idea of completing the missing details for visual content is reasonable. The method of synthetic discriminative training is interesting and straightforward for training the VPG-C without the need for supervised instructions.
- The proposed DEMON benchmark encompasses a wide range of tasks spanning multiple categories, offering the potential for the evaluation of future research efforts.

**Weaknesses:**

- The effectiveness of the proposed VPG-C has not been fully validated. On the proposed DEMON benchmark, the improvement of VPG-C compared to InstructBLIP is quite limited, which does not align with the expectation that completed details would significantly enhance VPG. The improvement on the MME benchmark is also not significant.
- The experimental evaluations on many common benchmarks are missing, such as the evaluation protocols of InstructBLIP (”divide the 26 datasets into 13 held-in datasets and 13 held-out datasets” including NoCaps, Flickr30K, GQA, VSR, IconQA, TextVQA, Visdial, HM, VizWiz, SciQA Image, MSVD QA, MSRVT QA, iVQA), the Mini-GPT4 dataset, the LLaVA-Instruct-150K benchmark, and the MIMIC-IT dataset. Evaluating on the same benchmarks as other MLLMs is important for a comprehensive and fair comparison.

**Questions:**

The primary concerns are related to experimental evaluation, and the rating would improve if the experimental evaluation were comprehensive.

---

> ### Author Response · Authors · 2023-11-18
> **Responses to Reviewer kFPK (Part I)**
>
> We sincerely appreciate your constructive and insightful comments. We will explain your concerns point by point.
>
> **Q1:** The effectiveness of the proposed VPG-C has not been fully validated. On the proposed DEMON benchmark, the improvement of VPG-C compared to InstructBLIP is quite limited, which does not align with the expectation that completed details would significantly enhance VPG. The improvement on the MME benchmark is also not significant.
>
> **A1:**  Thank you for your valuable feedback. We are grateful for the opportunity to further clarify and emphasize the unique strengths of our proposed VPG-C method.
>
> **(1.1) Data and Parameter Efficiency.** Our method requires fine-tuning only 6.3M parameters with 64K synthetic instruction data, which is significantly more efficient than InstructBLIP's approach of fine-tuning 107M parameters using 16M human-annotated instruction data [*Xu et al., 2023*].
>
> |              | Number of Tunable Parameters | Instruction Data Size | Training Data Type   |
> | :----------- | :--------------------------- | :-------------------- | :------------------- |
> | InstructBLIP | 107M                         | 16M                   | Human Annotation     |
> | VPG-C        | 6.3M                         | 64K                   | Synthetic Generation |
>
> **(1.2) Similarity in InstructBLIP's Training Data and Benchmark Tasks.** The InstructBLIP's training datasets share considerable similarity with tasks in the DEMON and MME benchmarks. For instance, the OCR-VQA dataset used in InstructBLIP is also employed in the DEMON benchmark. Further, tasks like "Visual Conversation," "OKVQA," and "Complex Reasoning" in InstructBLIP closely resemble "Multi-Modal Dialogue," "Knowledge Grounded QA," and "Multi-Image Reasoning" in the DEMON benchmark. A recent study [*Liu et al., 2023*] has also indicated that the type of instruction tuning data used in InstructBLIP can significantly enhance performance on benchmarks like MME, marking a considerable improvement from 502.8 to 1377.6.
>
> **(1.3) Further Experiments Inspired by Your Suggestions.** Motivated by your insightful suggestions, we conduct additional experiments to explore the scaling effect of instruction tuning data. We utilize a subset of InstructBLIP’s training data, including OKVQA, A-OKVQA, OCRVQA, TextCaps, and VQA-v2. As shown in the following tables, we find that even with a partial set of InstructBLIP's training data, the diversity and high quality of these manually annotated datasets substantially enhance the performance of VPG-C, notably surpassing InstructBLIP.
>
> |              | Multimodal Dialogue | Visual Storytelling | Visual Relation Inference | Multimodal  Cloze | Knowledge Grounded QA | Text-Rich  Images QA | Multi-Image Reasoning |
> | :----------- | :------------------ | :------------------ | :------------------------ | :---------------- | :-------------------- | :------------------- | :-------------------- |
> | InstructBLIP | 33.58               | 24.41               | 11.49                     | 21.20             | 47.40                 | 44.40                | 48.55                 |
> | VPG-C        | 37.50               | 25.20               | 25.90                     | 22.15             | 48.60                 | 44.93                | 50.28                 |
> | **VPG-C\***  | **39.68**           | **29.72**           | **26.31**                 | **25.83**         | **52.06**             | **48.65**            | **53.17**             |
>
> |              | MME-Perception | MME-Cognition |
> | :----------- | :------------- | :------------ |
> | InstructBLIP | 1212.82        | 291.79        |
> | VPG-C        | 1299.24        | 321.07        |
> | **VPG-C\***  | **1476.58**    | **392.47**    |
>
> *(\* indicates fine-tuning with InstructBLIP's training data)*
>
> We are grateful for your suggestions, which have significantly contributed to improving our paper! We have incorporated the results of these additional experiments into our revised manuscript (Appendix G) to provide a more comprehensive understanding of VPG-C's capabilities.

---

> > ### Author Response · Authors · 2023-11-18
> > **Responses to Reviewer kFPK (Part II)**
> >
> > **Q2:** The experimental evaluations on many common benchmarks are missing, such as the evaluation protocols of InstructBLIP (”divide the 26 datasets into 13 held-in datasets and 13 held-out datasets” including NoCaps, Flickr30K, GQA, VSR, IconQA, TextVQA, Visdial, HM, VizWiz, SciQA Image, MSVD QA, MSRVT QA, iVQA), the Mini-GPT4 dataset, the LLaVA-Instruct-150K benchmark, and the MIMIC-IT dataset. Evaluating on the same benchmarks as other MLLMs is important for a comprehensive and fair comparison.
> >
> > **A2:** Thank you for your constructive suggestions. As you nicely suggested, we have conducted additional experiments on these datasets to further evaluate the effectiveness of our method.
> >
> > **(2.1) Evaluation on InstructBLIP Benchmark.** Following the same setting as InstructBLIP, we evaluate our VPG-C on the 13 held-out datasets. As shown in the following table, with the enhanced  fine-grained visual perception capability, our method achieves the best zero-shot performance in 10 out of 13 datasets. This indicates that our VPG-C can consistently improve the performance across a wide variety of datasets.
> >
> > | **Method & Dataset**     | NoCaps    | Flickr30K | GQA      | VSR      | IconQA   | TextVQA  | Visdial  | HM       | VizWiz   | SciQA image | MSVD QA  | MSRVTT QA | iVQA     |
> > | :----------------------- | :-------- | :-------- | :------- | :------- | :------- | :------- | :------- | :------- | :------- | :---------- | :------- | :-------- | :------- |
> > | Flamingo-9B              | -         | 61.5      | -        | -        | -        | 31.8     | -        | 57.0     | 28.8     | -           | 30.2     | 13.7      | 35.2     |
> > | BLIP-2 (FlanT5XXL)       | 98.4      | 73.7      | 44.6     | 68.2     | 45.4     | 44.1     | 46.9     | 52.0     | 29.4     | 64.5        | 34.4     | 17.4      | 45.8     |
> > | BLIP-2 (Vicuna-7B)       | 107.5     | 74.9      | 38.6     | 50.0     | 39.7     | 40.1     | 44.9     | 50.6     | 25.3     | 53.8        | 18.3     | 9.2       | 27.5     |
> > | InstructBlip (FlanT5XXL) | 120.0     | 83.5      | 47.9     | 65.6     | 51.2     | 46.6     | **48.5** | 54.1     | 30.9     | 70.6        | 44.3     | 25.6      | 53.8     |
> > | InstructBlip (Vicuna-7B) | 123.1     | 82.4      | 49.2     | 54.3     | 43.1     | 50.1     | 45.2     | 59.6     | 34.5     | 60.5        | 41.8     | 22.1      | 52.2     |
> > | MiniGPT-4 (13B)          | -         | -         | 30.8     | 41.6     | 37.6     | -        | -        | -        | -        | -           | -        | -         | -        |
> > | LLaVA (13B)              | -         | -         | 41.3     | 51.2     | 43.0     | -        | -        | -        | -        | -           | -        | -         | -        |
> > | MiniGPT-v2               | -         | -         | 60.1     | 62.9     | 51.5     | -        | -        | 58.8     | 53.6     | -           | -        | -         | -        |
> > | LLaVA-1.5 (7B)           | -         | -         | **62.0** | -        | -        | 58.2     | -        | -        | 50.0     | 66.8        | -        | -         | -        |
> > | mPLUG-Owl2 (8.2B)        | -         | 85.1      | 56.1     | -        | -        | **58.2** | -        | -        | 54.5     | 68.7        | 42.4     | 23.6      | -        |
> > | VPG-C (Vicuna-7B)        | **123.5** | **86.2**  | 56.3     | **71.4** | **53.3** | 57.4     | 46.5     | **61.4** | **56.7** | **71.0**    | **47.4** | **28.9**  | **54.2** |

---

> ### Author Response · Authors · 2023-11-18
> **Responses to Reviewer kFPK (Part III)**
>
> **(2.2) GPT-4 Evaluation on MinGPT-4, LLaVA, MIMIC-IT Datasets.** We would like to respectfully clarify that the Mini-GPT4, LLaVA-Instruct-150K, and MIMIC-IT datasets are generated by LLMs and are primarily used for instruction tuning. These datasets do not have corresponding human-annotated test sets that are typically used for evaluation purposes. This unique nature of the datasets presents a challenge in terms of conventional evaluation protocols. Therefore, we leverage GPT-4 (text-only) to evaluate the quality of our model’s generated responses on these datasets.
>
> Specifically, we randomly sample 200 examples from each of these three datasets. The sampled instruction-response pairs, along with additional object bounding boxes and captions for the images, are input into GPT-4 as references. Following recent studies [*Liu et al., 2023; Yin et al, 2023*], we leverage GPT-4 to evaluate the helpfulness, relevance, accuracy, and level of details of the responses from MLLMs. GPT-4 provides an overall score on a scale from 1 to 10 for each response, where a higher score indicates better performance.
>
> As shown in the following table, we compare our VPG-C with InstructBLIP, MiniGPT-4, and LLaVA. While LLaVA and MiniGPT-4 have encountered some of the testing data during training, our VPG-C still achieves competitive performance. Particularly, our VPG-C demonstrates significant superiority on the challenging MIMIC-IT dataset, of which each instruction is accompanied by complex multi-modal in-context information.
>
> |              | MiniGPT-4 Dataset | LLaVA-150K | MIMIC-IT |
> | :----------- | :---------------- | :--------- | :------- |
> | LLaVA        | 7.2               | **8.5+**   | 5.2      |
> | MiniGPT-4    | **8.8+**          | 7.4        | 5.4      |
> | InstructBLIP | 7.3               | 7.5        | 6.5      |
> | **VPG-C**    | **8.1**           | **7.8**    | **7.6**  |
>
> *(+ indicates that the testing samples have been used during the training stage.)*
>
> **(2.3) GPT-4V Evaluation on MinGPT-4, LLaVA, MIMIC-IT Datasets.** To further illustrate the effectiveness of our method, we use recently opened GPT-4V, which can directly take images as input. Since the API interface is not available and there are strict limits on the number of uses, we randomly sample 30 examples from each dataset and manually prompt GPT-4V via web interface. We use similar prompt templates as a recent study [*Yin et al., 2023*]. The following table verifies the consistent superiority of our VPG-C on these complicated instruction data.
> |              | MiniGPT-4 Dataset | LLaVA-150K | MIMIC-IT |
> | :----------- | :---------------- | :--------- | :------- |
> | LLaVA        | 7.0               | **8.7+**   | 4.9      |
> | MiniGPT-4    | **8.4+**          | 7.3        | 5.0      |
> | InstructBLIP | 7.1               | 7.1        | 6.4      |
> | **VPG-C**    | **8.2**           | **7.6**    | **7.5**  |
>
> *(+ indicates that the testing samples have been used during the training stage.)*
>
> Thank you once again for your constructive suggestions, which have been instrumental in enhancing the quality of our research! We have now incorporated these experiments into our revised manuscript (Appendix H, I).
>
>
>
> **References**
>
> [*Xu et al., 2023*] LVLM-eHub: A Comprehensive Evaluation Benchmark for Large Vision-Language Models. Arxiv 2306.09265.
>
> [*Liu et al., 2023*] Improved Baselines with Visual Instruction Tuning. Arxiv 2310.03744.
>
> [*Yin et al., 2023*] Woodpecker: Hallucination Correction for Multimodal Large Language Models. Arxiv 2310.16045.

---

> > ### Comment · Reviewer_kFPK · 2023-11-20
> > **After Rebuttal**
> >
> > Thank you for your clarifications and further evaluations. My main concern regarding evaluation was addressed, so I will raise my score to a 6.

---

> > > ### Author Response · Authors · 2023-11-21
> > > **Thank you for your acknowledgement!**
> > >
> > > Thank you for raising the score. Your valuable suggestions greatly contribute to the quality of our manuscript. Thank you again for your precious time and valuable suggestions!

---

### Official Review · Reviewer_gX7w · 2023-10-31

**Soundness:** 3 good
**Presentation:** 3 good
**Contribution:** 3 good
**Rating:** 6
**Confidence:** 5

**Summary:**

This paper presents a new training method for Q-Former/Resampler, aiming to provide richer visual representations for lora-based Multimodal Language Models (MLLMs). In addition, the paper proposes a synthetic training dataset for training VPG-C. After training, VPG-C achieves surprising results on the benchmark proposed in this paper and other open-source benchmarks.

**Strengths:**

1. The authors are the first (at least to my knowledge) to propose using the latent features of the intermediate layers of the llm as guidance for the q-former, providing directed detail supplementation for the LLM.

2. The provided dataset/training method may inspire future research.

3. The provided benchmark can better diagnose the capabilities of MLLMs.

**Weaknesses:**

1. The paper does not mention the setting for instruction tuning. My understanding is that after using the synthetic discriminative training strategy, the model automatically acquires the ability to follow instructions without needing an instruction tuning phase.

2. The ablation in the paper validates the effectiveness of several proposed modules. But can VPG-C be applied to a finetuning setting, such as llava/minigpt4?

3. In multimodal dialogues, does the model need to update guidance multiple times when providing multiple answers? In other words, for each new additional question, does the model need to run the whole model to obtaion all hidden states and can't use the existing qkv cache?

**Questions:**

See weakness.

---

> ### Author Response · Authors · 2023-11-18
> **Response to Reviewer gX7w (Part I)**
>
> Thank you for your kind feedback and valuable comments. We have revised our manuscript and addressed several points you mentioned.
>
> **Q1:** The paper does not mention the setting for instruction tuning. My understanding is that after using the synthetic discriminative training strategy, the model automatically acquires the ability to follow instructions without needing an instruction tuning phase.
>
> **A1:** Thank you for your keen observation regarding our training approach. You are absolutely correct: the synthetic discriminative training strategy unleashes the model's instruction understanding ability, without further supervised instruction tuning stage.
>
> &nbsp;&nbsp;
>
> **Q2:** The ablation in the paper validates the effectiveness of several proposed modules. But can VPG-C be applied to a finetuning setting, such as llava/minigpt4?
>
> **A2:** Thank you for the constructive suggestion. As you nicely suggested, we adopt our VPG-C module to fine-tune LLaVA and MiniGPT-4 and report the results as follows.
>
> **(2.1) Evaluation on DEMON Benchmark.** As shown in the following table, our method can be well adapted to LLaVA and MiniGPT-4 in a fine-tuning setting, empowering them to effectively attend to fine-grained details. For instance, on Visual Relation Inference, VPG-C brings about an improvement of 10.91 and 13.47 points for LLaVA and MiniGPT-4, respectively. Also, on Multimodal Dialogue, VPG-C largely improves LLaVA and MiniGPT-4 by 9.22 and 8.2 points, respectively. These results indicate that VPG-C enables them to effectively associate the interleaved images mentioned in the complicated context.
>
> |                       | Multimodal Dialogue | Visual Storytelling | Visual Relation Inference | Multimodal  Cloze | Knowledge Grounded QA | Text-Rich  Images QA | Multi-Image Reasoning |
> | :-------------------- | :------------------ | :------------------ | :------------------------ | :---------------- | :-------------------- | :------------------- | :-------------------- |
> | LLaVA                 | 7.79                | 10.70               | 8.27                      | 15.85             | 36.20                 | 28.33                | 41.53                 |
> | **LLaVA + VPG-C**     | **17.01**           | **18.37**           | **19.18**                 | **19.07**         | **43.22**             | **35.43**            | **47.42**             |
> | MiniGPT-4             | 13.69               | 17.07               | 7.95                      | 16.60             | 30.27                 | 26.40                | 43.50                 |
> | **MiniGPT-4 + VPG-C** | **21.89**           | **26.80**           | **21.42**                 | **24.53**         | **39.67**             | **34.36**            | **50.72**             |
>
> **(2.2) Evaluation on MME Benchmark**. Furthermore, we report the results on MME benchmark. The following table validates that our VPG-C is also beneficial to the perception and cognition abilities of LLaVA and MiniGPT-4.
>
> |                       | MME-Perception | MME-Cognition |
> | :-------------------- | :------------- | :------------ |
> | LLaVA                 | 502.82         | 214.64        |
> | **LLaVA + VPG-C**     | **806.32**     | **286.53**    |
> | MiniGPT-4             | 866.57         | 292.14        |
> | **MiniGPT-4 + VPG-C** | **1107.20**    | **342.65**    |
>
> Based on the above experiments, we believe that VPG-C is a reliable fine-tuning method to enhance the existing MLLMs' perception and reasoning abilities for visual inputs. Thank you once again for your valuable suggestions! We have now incorporated these experiments into our revised manuscript (Appendix F).

---

> ### Author Response · Authors · 2023-11-18
> **Response to Reviewer gX7w (Part II)**
>
> **Q3:** In multimodal dialogues, does the model need to update guidance multiple times when providing multiple answers? In other words, for each new additional question, does the model need to run the whole model to obtaion all hidden states and can't use the existing qkv cache?
>
> **A3:** Thank you for your insightful query. In our experiments, we update the guidance for each new question. Technically, it's feasible to reuse guidance generated from previous questions, allowing the model to use the existing qkv cache efficiently. This introduces minimal extra time overhead. However, each question in a dialogue may focus on different aspects of the image, making it sometimes necessary to reintegrate new residual details for a new question. Regenerating guidance for each question could theoretically improve response reliability but at the cost of increased time overhead.
>
> In the following table, we present a comparison of the average test time and performance across each dataset in multimodal dialogue tasks within DEMON benchmark. We can see that:
>
> - when regenerating guidance for each new additional question, the extra time overhead might not be significant (the test time is still less than that of the LLaVA model). Moreover, this approach does indeed achieve higher performance compared to other methods.
>
> - even when reusing previously generated guidance for each new question, our method still outperforms other baselines in terms of performance, while maintaining a similar level of time expenditure.
>
> Overall, the decision of whether to regenerate guidance for each new question in multi-modal dialogue is flexible and can be user-determined, based on the specific requirements and context of the scenario.
>
> | Method                          | AVG test time | performance |
> | :------------------------------ | :------------ | :---------- |
> | MiniGPT4                        | 655s          | 13.69       |
> | LLaVA                           | 1284s         | 7.79        |
> | VPG-C w/o guidance regeneration | 676s          | 34.66       |
> | VPG-C w/ guidance regeneration  | 820s          | **37.50**   |

---

> ### Comment · Reviewer_gX7w · 2023-11-22
>
> Thanks for the authors’ efforts in rebuttal. I keep my score as it is.

---

> > ### Author Response · Authors · 2023-11-22
> > **Thank you for your acknowledgement!**
> >
> > Thank you again for your precious time and valuable suggestions.

---

### Official Review · Reviewer_cxqK · 2023-11-03

**Soundness:** 3 good
**Presentation:** 3 good
**Contribution:** 3 good
**Rating:** 8
**Confidence:** 4

**Summary:**

This paper studies the instruction tuning in multimodal language models. In particular, it tries to improve the bottleneck of visual prompt generator (VPG), aka the visual feature converter which converts a generic visual embedding into LLM-interpretable inputs. It hypothesizes that the bottleneck comes from the lacking of attention to details in VPG. To address this, the paper proposed two components: (1) a VGP-C architecture which additionally generate features for intermediate LLM layers with attention to LLM intermediate features, (2) a synthetic data generation procedure to generate training data to teach VPG to attend to details. In addition to these, the paper also introduced a new evaluation benchmark.

**Strengths:**

1. The paper proposed two methods for multimodal instruction following, the VPGC architecture and the data synthesis technique. Both of them are novel and inspiring.
2. Plenty of ablation studies are provided to support the effectiveness of the method. Comprehensive experiments also demonstrate the superiority of the method compared to existing models.
3. The paper also introduced a benchmark for future research.

**Weaknesses:**

1. The method contains several steps and is thus quite complicated. It may be hard to reproduce the whole framework in different settings and code bases.

**Questions:**

NA.

---

> ### Author Response · Authors · 2023-11-18
> **Response to Reviewer cxqK**
>
> We sincerely thank you for your comprehensive comments and constructive advice. We will explain your concern as follows.
>
> **Q1:** The method contains several steps and is thus quite complicated. It may be hard to reproduce the whole framework in different settings and code bases.
>
> **A1:**  We apologize if our manuscript gave the impression that the implementation of VPG-C is complex. Allow us to clarify some key points that we hope will address your concerns.
>
> **(1.1) Simplicity and Compatibility of VPG-C.** Contrary to the perceived complexity, the implementation of VPG-C is quite straightforward. It does not require any modifications to the internal structures of either the VPG or the LLM. VPG-C performs like a prompt tuning approach at the intermediate layer of an LLM. Furthermore, as demonstrated in our experiments in Section 4.4 (specifically Table 4 Rows 7, 8, and 9), VPG-C is compatible with different VPGs and LLM backbones, and achieves performance improvements across these variations. This underscores the method's simplicity and adaptability across different models.
>
> **(1.2) Extended Experiments.** To further support our claim, we have included additional experimental results in the following table. We apply our method to fine-tune LLaVA and MiniGPT-4, which are implemented with different code bases. The results show consistent performance improvements, indicating the adaptability and effectiveness of VPG-C across various settings and code bases.
> |                       | Multimodal Dialogue | Visual Storytelling | Visual Relation Inference | Multimodal  Cloze | Knowledge Grounded QA | Text-Rich  Images QA | Multi-Image Reasoning |
> | :-------------------- | :------------------ | :------------------ | :------------------------ | :---------------- | :-------------------- | :------------------- | :-------------------- |
> | LLaVA                 | 7.79                | 10.70               | 8.27                      | 15.85             | 36.20                 | 28.33                | 41.53                 |
> | **LLaVA + VPG-C**     | **17.01**           | **18.37**           | **19.18**                 | **19.07**         | **43.22**             | **35.43**            | **47.42**             |
> | MiniGPT-4             | 13.69               | 17.07               | 7.95                      | 16.60             | 30.27                 | 26.40                | 43.50                 |
> | **MiniGPT-4 + VPG-C** | **21.89**           | **26.80**           | **21.42**                 | **24.53**         | **39.67**             | **34.36**            | **50.72**             |
>
> **(1.3) Future Improvements.** We greatly appreciate your suggestion and will optimize our open-source project and documents. We recognize the importance of reproducibility and ease of use in the research community. In response, we are committed to enhancing our code repository with more detailed documentation and examples, making it more accessible for researchers and practitioners who wish to apply our method in different contexts.
>
> We hope this explanation clarifies the simplicity and practicality of our method. We are grateful for the opportunity to improve our work based on your insightful comments and look forward to further discussions.

---

### Official Review · Reviewer_HGkc · 2023-11-10

**Soundness:** 3 good
**Presentation:** 4 excellent
**Contribution:** 3 good
**Rating:** 8
**Confidence:** 4

**Summary:**

This paper aims to improve reasoning capabilities of Multi-modal Large Language Models (MLLMs) for demonstrative instructions. The authors highlight that most of MLLMs have been over-optimized on the image-captioning objective which has led to the use of visual features that could only describe the captions while neglecting its focus on minor yet discriminative features important for fine-grained reasoning.

Firstly, this work proposes VPG-C which is a lightweight adaption on top of VPG which aims to reuse the information from the intermediate LLM layer and is used to modulate the VPG features through guidance. The modified VPG features are integrated into LLM intermediate layer which effectively improves the fine-grained reasoning performance.

Secondly, to train the VPG-module, this work proposes a automatic way to generate synthetic data used to improve fine-grained discriminative performance of MLLMs.

Lastly, DEMON benchmark is proposed to evaluate the demonstrative instruction understanding of the proposed technique and other MLLMs.

The proposed approach is fairly motivated with analysis and ablation studies.

**Strengths:**

Strengths:

1) This paper identifies and aims to address a crucial limitation of lack of reasoning capabilities of Multi-modal Large Language Models (MLLMs) for demonstrative instructions. Improving MLLMs for demonstrative instructions will pave more rapid growth for building human-friendly AI assistants.

2) The proposed VPG-C design is fairly motivated and it is compute friendly.

3) The idea of generating synthetic data with automatic pipeline to improve fine-grained discriminative capabilities of MLLMs is encouraging.

4) The authors have proposed a suitable benchmark demon which would enable more systematic developments in improving MLLMs.

**Weaknesses:**

I could not observe any significant weaknesses. However I have a concern regarding the inference compute efficiency of the proposed approach.

1) As the model is reusing its intermediate features via a feedback loop system, this will significantly increase the training and testing time and might not be batch friendly during inference. How does the throughput of this technique compares against previous methods?

**Questions:**

Please refer to weakness section.

---

> ### Author Response · Authors · 2023-11-18
> **Response to Reviewer HGkc**
>
> We sincerely thank you for the valuable comments. We are encouraged to see that our work is recognized as fairly motivated and encouraging. We will explain your concern as follows.
>
> **Q1:** As the model is reusing its intermediate features via a feedback loop system, this will significantly increase the training and testing time and might not be batch friendly during inference. How does the throughput of this technique compares against previous methods?
>
> **A1:** Thank you for raising an important concern. Our model derives the guidance from the intermediate layer of the LLM and seamlessly integrates the missing visual details into the subsequent layer via a skip connection. Therefore, our model does not introduce any additional LLM forward propagation processes and only requires an additional call of the VPG. Concerning **time overhead**, the forward propagation time of our 107M VPG is considerably less than that of a 7B LLM. As a result, the additional call to VPG does not significantly increase the overall training or testing time. Regarding **memory usage**, we effectively reuse the original VPG and the initial visual features generated by the VIT. VPG-C introduces minimal new model parameters (*i.e.*, a set of query embeddings and two linear projection layers). Hence, the additional memory overhead is minimal.
>
> Next, we will separately compare the throughput of VPG-C with MiniGPT4, InstructBLIP, and LLaVA during both the training and inference stages.
>
> **(1.1) Training Efficiency.** Despite the additional VPG call, the overall impact on training time is mitigated by its inherently lightweight design. Since our proposed VPG-C module is lightweight with fewer trainable parameters (6.3M) compared to InstructBLIP (fine-tuning the Q-former) and LLaVA (fine-tuning the LLM), VPG-C requires less time overhead during the stages of backward propagation and parameter updating. To validate the above viewpoint, we use the same image-text pairs to train some MLLMs with a batch size of 48, ensuring identical hyperparameters. As shown in the following table, the throughput of VPG-C during training is **slightly lower than MiniGPT4 but higher than InstructBLIP (and LLaVA).**
>
> | Method       | Trainable Parameters | AVG Training Time per Iteration |
> | :----------- | :-------------------- | :------------------------------ |
> | MiniGPT-4    | 3.1M                  | 1.04s / iter                    |
> | InstructBLIP | 107M                  | 1.25s / iter                    |
> | LLaVa        | 7B                    | &gt;2.5s / iter*                |
> | VPG-C        | 6.3M                  | 1.09s / iter                    |
>
> *(\*Estimated, considering LLaVA model requires fine-tuning of the whole LLM, its training time per iteration is significantly higher than that of the other three models.)*
>
> **(1.2) Inference Efficiency.** Our tests using DEMON benchmark samples reveal that VPG-C only marginally extends the generation time (by approximately 6.3%).  As shown in the following table, VPG-C's throughput during inference is competitive, only **slightly lower than MiniGPT4 but higher than both InstructBLIP and LLaVA**. Importantly, its GPU memory consumption is only marginally increased, ensuring batch efficiency.
>
> | Method       | AVG number of responses generated per second | AVG GPU memory usage |
> | :----------- | :--------------------------------------- | :------------------- |
> | MiniGPT-4    | 12.6                                     | 32G                  |
> | InstructBLIP | 10.3                                     | 43G                  |
> | LLaVa        | 7.0                                      | 52G                  |
> | VPG-C        | 11.8                                     | 37G                  |
>
> In summary, while our model introduces a modest increase in processing time, it remains efficient and batch-friendly, particularly when considering its enhanced capabilities in completing visual details into MLLMs. We believe these trade-offs are justified given the model's performance gains.
>
> Thank you once again for your constructive suggestion, which has been instrumental in enhancing the quality of our research! We have now incorporated the efficiency analysis into our revised manuscript (Appendix E).

---

> > ### Comment · Reviewer_HGkc · 2023-11-18
> >
> > Thank you for providing a detailed rebuttal!
> >
> > Yes, given the performance gains and significance of the problem addressed in this paper, the current overhead should not be an issue and future works can always improve the work in this direction.
> >
> > Overall I am satisfied with this work and will keep my rating as it is.

---

> > > ### Author Response · Authors · 2023-11-19
> > > **Thank you for your acknowledgement!**
> > >
> > > Thank you for your precious time and continued support. We appreciate your recognition of our work's significance.

---

### Author Response · Authors · 2023-11-18
**General Response to All Reviewers**

We sincerely thank all the reviewers for their insightful and valuable comments! Overall, we are encouraged that they find that:

- The proposed lightweight VPG-C architecture and the synthetic training strategy are both novel and inspiring (all reviewers).
- The proposed comprehensive benchmark is beneficial to future research (all reviewers).
- The proposed method is fairly motivated with plenty of analysis and ablation studies (Reviewer HGkc, Reviewer cxqK).
- Improving MLLMs for demonstrative instructions will pave more rapid growth for building human-friendly AI assistants (Reviewer HGkc).

We have revised the manuscript according to the reviewers' comments (the changes have been highlighted in blue font). The main updates are summarized as follows:

- In Appendix E, we systematically discuss the training and inference efficiency of our model and compare it with several state-of-the-art models.
- In Appendix F, we add experiments of applying VPG-C to fine-tune  LLaVA and MiniGPT-4.
- In Appendix G, we study the scaling effect of instruction tuning data.
- In Appendix H, we add experiments on the InstructBLIP benchmark.
- In Appendix I, we add GPT-4 evaluation on the MiniGPT-4, LLaVA, and MIMIC-IT training datasets.

Next, we address each reviewer's detailed concerns point by point. We hope we have addressed all of your concerns. Discussions are always open. Thank you!

---

### Meta-Review · Area_Chair_S33P · 2023-12-09

**Metareview:**

This paper seeks to enhance the reasoning abilities of Multi-modal Large Language Models (MLLMs) concerning demonstrative instructions, motivated by shortage of existing visual prompt generators usually concentrate solely on primary visual contents only. To solve the problem, this work proposes a new module VPG-C, and a synthetic discrimitive strategy to train VPG-C. During evaluation, a new benchmark DEMON is built for demonstrative instruction understanding.

On the pro side, the reviewers agree:
1. The proposed new VPG-C module and corresponding training strategy is novel.
2. The proposed benchmark is a good contribution and could enable more development of MLLMs in the community.
3. The experiments and restyle are strong and convincing.

On the con side, the reviewers raised concerns about:
1. Inference computation efficiency.
2. Reproducibility

During the rebuttal phase, the authors responded to these concerns by providing additional details and conducting additional experiments. Furthermore, they illustrated the reliability of the VPG-C component and its applicability to other existing MLLM models, such as MiniGPT4 and LLaVA.

In conclusion, reviewers express positivity towards the paper, acknowledging its effective addressing of a valid problem and finding the results to be convincing.

**Justification For Why Not Higher Score:**

See the comment above.

**Justification For Why Not Lower Score:**

The paper got a average rating of 7, with all reviewers are positive about the paper.

---

### Decision · Program_Chairs · 2024-01-16

Accept (spotlight)